# Non-Asymptotic Analysis of Median-of-Means Estimation for High-Dimensional Time Series

Haotian Xu

Department of Mathematics and Statistics, Auburn University

Dan Luo

Agios Pharmaceuticals

Stéphane Guerrier

Faculty of Science, University of Geneva

Runze Li

Department of Statistics, Pennsylvania State University

Yuan Ke*

Department of Statistics, University of Georgia

## Abstract

This study addresses the challenges in estimating mean vectors and autocovariance matrices in modern data settings, which are often affected by three key issues: high-dimensionality, heavy-tailed distributions, and temporal dependence. These salient features lead to reduced performance of many existing methods. To tackle these challenges, we introduce a computationally efficient framework centered around the median-of-means. In particular, we study the non-asymptotic properties of median-of-means estimators for mean vectors and autocovariance matrices for high-dimensional, heavy-tailed dependent data. Extensive simulations demonstrate the finite sample and computational advantages of the proposed estimators. The effectiveness of our approach is further validated through its application in analyzing the autocovariance patterns of S&P 500 daily returns over the past two decades.

**Keywords:** Autocovariance matrix, Financial time series analysis, Heavy-tailed distribution, Outlier resilience, Tail-robust statistical method

**Mathematics Subject Classification (2020):** 62M10

## 1 Introduction

Advances in data collection technology have led to rapid growth in big data applications, often involving simultaneous measurement of hundreds or thousands of time series. However, the high-dimensionality of these datasets presents significant challenges for time series analysis. These datasets often contain asymmetric and heavy-tailed time series that deviate from the sub-Gaussian assumption. As a result, standard estimation methods that perform well in

---

*Corresponding author: yuan.ke@uga.edu

low-dimensional settings often fail to deliver reliable results in high-dimensional contexts. To overcome these challenges, it is essential to develop estimators that are robust against outliers and heavy-tailedness while exhibiting sub-Gaussian behavior. Although research focuses primarily on tail-robust mean and covariance estimation for independent data (Catoni, 2012; Nemirovsky and Yudin, 1983; Lerasle and Oliveira, 2011; Lugosi and Mendelson, 2019; Avella-Medina et al., 2018), there is a growing need for tail-robust estimation methods that can handle the temporal dependence of time series data. Using the framework of functional dependence introduced by Wu (2005), Zhang (2021) studied tail-robust estimation of high-dimensional mean vectors and covariance matrices using robust $M$-estimators. More recently, Wang and Tsay (2023) established rate-optimal robust estimators for several structured high-dimensional vector autoregressive models, via a constrained Yule–Walker step combined with a truncation-based robust autocovariance estimator, under a bounded $(2 + 2\epsilon)$-th moment ($\epsilon \in (0, 1]$) and a geometrically decaying $\alpha$- (or $\beta$-) mixing condition with coefficients of order $O(r^\ell)$, $r \in (0, 1)$. Their endpoint $\epsilon = 1$ is precisely the finite-fourth-moment condition under which we obtain the optimal rate for autocovariance estimation (equivalently, a finite variance of the lag-products), while their $\epsilon \in (0, 1)$ further covers the sub-fourth-moment regime at a correspondingly slower rate. Our contribution is complementary along the remaining axes: we estimate the mean vector and autocovariance matrix directly under a functional dependence measure assumption rather than the parameters of a structured VAR, and, for the low-dimensional mean, our theory permits genuinely *polynomially* decaying functional dependence rather than the exponentially decaying of temporal dependence. In fields such as financial market analysis, climate modeling, and sensor networks, ignoring temporal dependence can exacerbate the impact of outliers and heavy-tailed distributions, leading to suboptimal estimates. Tail-robust methods that account for this temporal structure are essential to provide reliable insights and improve decision-making in these high-stakes, data-rich environments.

The median-of-means (MoM) estimator has gained attention as a tail-robust method for mean estimation in the presence of outliers and heavy-tailed distributions. Initially introduced in Nemirovsky and Yudin (1983) and possibly explored in earlier works, the MoM method involves partitioning a sample into multiple subsets, calculating the mean of each subset, and then taking the median of these means as the final estimate. By leveraging the robustness of the median, this approach reduces the impact of outliers and heavy-tailed distributions on the traditional sample mean estimation. Additionally, the MoM method is computationally efficient compared to other tail-robust methods such as Huber-type regressions (see e.g., Minsker, 2018; Ke et al., 2019; Sun et al., 2020) which typically involve iterative optimization procedures. The simplicity and speed of the MoM method arise from its straightforward computations, which are easily implementable using parallel computing tools. Indeed, efficient algorithmic implementations have enhanced its practical use in large-scale data scenarios typical of modern statistical learning (see e.g., Arnaudon et al., 2013; Zhang and Liu, 2021; Pan and Owen, 2024).

Recent advancements have extended the application of MoM to high-dimensional and/or data contamination settings, further demonstrating its robustness and versatility. For example, Minsker (2015) proposed a permutation-invariant MoM that performs optimally under weakened moment conditions. Similarly, Lerasle et al. (2019) investigated robust mean embedding

estimation in reproducing kernel Hilbert spaces, showing that the MoM estimator achieves sub-Gaussian deviation bounds with minimal assumptions. The authors of Lecué and Lerasle (2020) applied the MoM method to machine learning, highlighting its robustness and efficiency in handling data contamination. More recently, Humbert et al. (2022) extended MoM method to kernel density estimation, proposing a robust method that efficiently handles outliers and heavy-tailed data. These developments underscore the continued progress in MoM estimation methods and their capability to handle modern, complex datasets while maintaining robustness and computational efficiency.

Despite the popularity of the MoM method, its existing non-asymptotic guarantees are largely confined to independent data; to the best of our knowledge, no non-asymptotic theory is available for the MoM estimator of high-dimensional time series under the functional (physical) dependence measure. This paper closes that gap. We establish non-asymptotic max-norm error bounds for the MoM estimator of the high-dimensional *mean vector*—Theorem 3.1 for the low-dimensional regime under temporal dependence allowed to decay only *polynomially* (the DAN condition), and Theorem 3.2 for the high-dimensional regime ($\log d = o(n)$) under a geometric-moment-contraction (GMC) condition—and for the lag-*l* *autocovariance matrix* (Theorem 3.3, under GMC). These results require only weak moment conditions, namely a finite *second* moment for the mean and a finite *fourth* moment for the autocovariance, and the high-dimensional bounds attain the minimax-optimal rate. We further demonstrate through extensive simulations that the MoM estimator delivers substantial finite-sample improvements over the sample mean and sample autocovariance under heavy-tailed and contaminated data, while remaining computationally efficient relative to iterative tail-robust alternatives. We illustrate the benefits of the MoM estimator for autocovariance change point detection in high-dimensional financial data. Consequently, the MoM estimation approach provides a theoretically sound framework for analyzing high-dimensional time series, addressing some important challenges posed by modern big data applications.

The remainder of this paper is organized as follows. Section 2 reviews essential concepts in high-dimensional time series analysis. Section 3 introduces the proposed MoM estimation methods and their theoretical properties. Section 4 presents simulation experiments comparing the proposed methods with existing approaches. Section 5 applies the proposed methods to a real-world high-dimensional time series dataset. Finally, Section 6 concludes the paper.

## 1.1 Notation

This subsection introduces the mathematical notations used throughout this paper. Let $\mathbb{Z}$, $\mathbb{N}^+$, and $\mathbb{R}$ denote the set of integers, natural numbers without 0, and real numbers, respectively. For a set $\mathbb{A}$, $\text{card}(\mathbb{A})$ denotes the cardinality of $\mathbb{A}$. For $n \in \mathbb{N}^+$, denote $[n] := \{1, 2, \ldots, n\}$. The superscript $\intercal$ denotes the transpose for matrices or vectors. Given a vector $\boldsymbol{x} = (x_1, \ldots, x_d)^\intercal \in \mathbb{R}^d$, the vector $l_q$-norm is written as $|\boldsymbol{x}|_q := \left(\sum_{j=1}^d |x_j|^q\right)^{1/q}$ for $1 \leq q < \infty$ and the vector $l_\infty$-norm is written as $|\boldsymbol{x}|_\infty := \max_{j \in [d]} |x_j|$. Further, we define $\|\boldsymbol{x}\|_q = \left(\sum_{j=1}^d \mathbb{E}|x_j|^q\right)^{1/q}$ for $1 \leq q < \infty$. Given a matrix $\mathbf{A} = (A_{kl})_{k \in [d_1]; l \in [d_2]} \in \mathbb{R}^{d_1 \times d_2}$, if $d_1 = d_2 = d$, $\text{tr}(\mathbf{A})$ and $\det(\mathbf{A})$ denote the trace and the determinant of $\mathbf{A}$, respectively. If $\mathbf{A}$ is a symmetric matrix, $\lambda_{\max}(\mathbf{A})$ and $\lambda_{\min}(\mathbf{A})$ denote the largest and smallest eigenvalues of $\mathbf{A}$, respectively. The spectral-norm,

the Frobenius-norm, the 1-norm, the $\infty$-norm and the max-norm of $\mathbf{A}$ are respectively denoted as $\|\mathbf{A}\| := \sqrt{\lambda_{\max}(\mathbf{A}^\intercal\mathbf{A})}$, $\|\mathbf{A}\|_\mathrm{F} := \sqrt{\mathrm{tr}(\mathbf{A}^\intercal\mathbf{A})}$, $\|\mathbf{A}\|_1 := \max_l \sum_{k=1}^d |A_{kl}|$, $\|\mathbf{A}\|_\infty := \max_k \sum_{l=1}^d |A_{kl}|$ and $\|\mathbf{A}\|_{\max} := \max_{k,l} |A_{kl}|$. For a sequence of matrices $\{\mathbf{A}_i\}_{i\in S}$ with $S \subseteq \mathbb{Z}$, we write $A_{i,(kl)}$ as the $(k,l)$-th entry of $\mathbf{A}_i$. $\mathbf{I}_d$ denotes the $d$-dimensional identity matrix. For an $\mathbb{R}$-valued random variable $X$, $\mathrm{kurt}(X) := \mathbb{E}(X - \mu)^4/\sigma^4$ denotes the kurtosis of $X$, where $\mu := \mathbb{E}X$ and $\sigma^2 := \mathrm{var}(X)$. For $q > 0$, we write the $L_q$-norm of $X$ as $\|X\|_q := \left(\mathbb{E}|X|^q\right)^{1/q}$. For $a \in \mathbb{R}$, let $\lfloor a \rfloor := \max\{z \in \mathbb{Z}, z \le a\}$. For $a, b \in \mathbb{R}$, we denote the sign of $a$ by $\mathrm{sign}(a)$, and denote $a \wedge b := \min(a, b)$ and $a \vee b := \max(a, b)$. For two positive values $a$ and $b$, we write $a \asymp b$ (resp. $a \lesssim b$) if there exist absolute constants $C_2 \ge C_1 > 0$ such that $C_1 \le a/b \le C_2$ (resp. $a/b \le C_1$). Absolute constants are denoted as $C, C_1, C_2, \cdots > 0$, which may be different in each place.

## 2 Preliminaries

In this section, we review some key concepts to characterize high-dimensional time series, which lay the groundwork for the non-asymptotic analysis of MoM estimator.

### 2.1 High-Dimensional Stationary Time Series

This paper focuses on high-dimensional stationary time series, specifically vector time series that consist of multiple univariate time series. Following the notation of Zhang and Wu (2017), a stationary $\mathbb{R}^d$-valued time series $\{\mathbf{X}_i\}_{i\in\mathbb{Z}}$ is defined as

$$\mathbf{X}_i = (X_{i,1}, X_{i,2}, \ldots, X_{i,d})^\intercal = G(\mathcal{F}_i), \tag{2.1}$$

where $G(\cdot) = (g_1(\cdot), g_2(\cdot), \ldots, g_d(\cdot))^\intercal$ is an $\mathbb{R}^d$-valued measurable function, and $\mathcal{F}_i = (\ldots, \epsilon_{i-1}, \epsilon_i)$ is a filtration with $\{\epsilon_i\}_{i\in\mathbb{Z}}$ being a sequence of i.i.d. random elements. Here, $G(\cdot)$ is independent of the time index $i$, ensuring that $\{\mathbf{X}_i\}_{i\in\mathbb{Z}}$ is stationary. This representation can be extended to locally stationary or non-stationary processes by allowing $G$ to depend on $i$ (Dahlhaus et al., 2019; Zhou and Wu, 2009).

The representation in (2.1) encompasses a wide class of stationary processes (Wu, 2005, 2011) and is frequently used to develop statistical tools under dependence. For instance, Wu (2005) introduced the functional dependence measure to quantify nonlinear dependence in terms of moments for stationary processes. Compared to other dependence measures, such as strong mixing coefficients (Dedecker et al., 2007; Bradley, 2007), the causal representation in (2.1) has several advantages. For example, it simplifies the verification of dependence conditions for complex nonlinear processes. Given a process in the form of (2.1), one can easily construct a martingale approximation (Wu, 2007) or an $m$-dependence approximation (Berkes et al., 2009; Liu et al., 2013) using coupling techniques (Berkes et al., 2009). Errors in these approximations can then be quantified using established theories for martingale difference sequences or i.i.d. sequences.

## 2.2 Functional Dependence Measure

In this subsection, we outline conditions for quantifying temporal dependence in a stationary process $\{\mathbf{X}_i\}_{i\in\mathbb{Z}}$ of the form (2.1). Let $\mathbf{X}_i' := G(\mathcal{F}_i')$ represent a coupled version of $\mathbf{X}_i$, where $\mathcal{F}_i' = (\ldots, \epsilon_{-1}, \epsilon_0', \epsilon_1, \ldots, \epsilon_i)$ is derived by replacing $\epsilon_0$ with an independent copy $\epsilon_0'$. Here, $\{\epsilon_j'\}_{j\in\mathbb{Z}}$ is a sequence of i.i.d. random variables, independent of $\{\epsilon_i\}_{i\in\mathbb{Z}}$. This construction quantifies the effect of perturbing $\epsilon_0$.

For some $q > 0$, assume the moment condition $\max_{j\in[d]} \|X_{i,j}\|_q < \infty$. The coordinate-wise functional dependence measure of order $q$ is defined for $i \in \mathbb{Z}$ and $j \in [d]$ as:

$$\delta_{i,q,j} := \|X_{i,j} - X_{i,j}'\|_q = \|g_j(\mathcal{F}_i) - g_j(\mathcal{F}_i')\|_q,$$

which captures the expected perturbation resulting from replacing $\epsilon_0$. For time series with short-range dependence, $\delta_{i,q,j}$ decays quickly to 0 as the time lag $i$ increases. Two commonly considered decay rates are (i) polynomial decay and (ii) exponential decay.

Following Wu and Wu (2016), we define the coordinate-wise Dependence Adjusted Norm (DAN) of order $q$ as:

$$\|X_{.,j}\|_{q,\nu} := \sup_{m\geq 0}(m+1)^\nu \Delta_{m,q,j}, \text{ for some } \nu \geq 0, \tag{2.2}$$

where $\Delta_{m,q,j} := \sum_{i=m}^{\infty} \delta_{i,q,j}$ represents the cumulative dependence measure. If $\|X_{.,j}\|_{q,\nu} < \infty$, the time series $\{X_{i,j}\}_{i\in\mathbb{Z}}$ exhibits polynomial decay, and we say the series is DAN($q$).

In Wu and Wu (2016), the dependence adjusted sub-Gaussian and sub-Exponential norms are defined as:

$$\|X_{.,j}\|_{\psi_\alpha,\nu} := \sup_{q\geq 2} q^{-1/\alpha}\|X_{.,j}\|_{q,\nu}, \quad \text{for } \alpha = 2 \text{ (sub-Gaussian) or } \alpha = 1 \text{ (sub-Exponential)}.$$

These norms require that $\|X_{.,j}\|_{q,\nu}$ grows slower than $Cq^{-1/\alpha}$, implying the existence of finite exponential moments in the marginal distributions. However, in this paper, we do not impose such strong conditions.

Next, we define the coordinate-wise Geometric Moment Contraction (GMC) of order $q$ as:

$$\|X_{.,j}\|_q := \sup_{m\geq 0} \rho^{-m}\Delta_{m,q,j}, \quad \text{for some } \rho \in (0,1). \tag{2.3}$$

If $\|X_{.,j}\|_q < \infty$, the series $\{X_{i,j}\}_{i\in\mathbb{Z}}$ exhibits exponential decay and is said to be GMC($q$). Many commonly used models satisfy these conditions: stationary causal ARMA, (G)ARCH, threshold and bilinear autoregressions, and iterated random function (Markov) models are GMC($q$) under standard parameter conditions, whereas linear processes with polynomially decaying coefficients are DAN($q$) but in general not GMC($q$); see Wu (2005, 2011) for explicit functional dependence measure calculations.

To address the high-dimensionality of $\{\mathbf{X}_i\}_{i\in\mathbb{Z}}$, we aggregate the coordinate-wise quantities in (2.2) and (2.3) using the $l_\infty$-norm:

$$\|X_.\|_{q,\nu} := \max_{j\in[d]}\|X_{.,j}\|_{q,\nu}, \quad \|X_.\|_q := \max_{j\in[d]}\|X_{.,j}\|_q.$$

The DAN and GMC norms allow us to impose temporal dependence conditions based on the decay rate of $\delta_{i,q,j}$. The following lemma demonstrates the "downward-compatibility" property of DAN and GMC norms. That is, $\text{GMC}(q_1)$ (or $\text{DAN}(q_1)$) implies $\text{GMC}(q_2)$ (or $\text{DAN}(q_2)$) for any $q_2 \in (0, q_1)$.

**Lemma 2.1.** *Let $q_1 > 0$ and $q_2 \in (0, q_1)$.*

(a) *If $\{X_{i,j}\}_{i \in \mathbb{Z}}$ is DAN($q_1$) for some $\nu \geq 0$, then it is also DAN($q_2$) for the same $\nu$.*

(b) *If $\{X_{i,j}\}_{i \in \mathbb{Z}}$ is GMC($q_1$) for some $\rho \in (0, 1)$, then it is also GMC($q_2$) for the same $\rho$.*

The GMC norm also possesses an "equivalence" property, as shown in Lemma 2 of Wu and Min (2005). Specifically, for $\{X_{i,j}\}_{i \in \mathbb{Z}}$ with $\|X_{i,j}\|_q < \infty$, if $\text{GMC}(q_1)$ holds for some $q_1 \in (0, q)$, then $\text{GMC}(q_2)$ holds for all $q_2 \in (0, q)$. However, this property does not hold for the DAN norm.

# 3 Median-of-Means Estimation for High-dimensional Time Series

In this section, we develop a non-asymptotic framework for analyzing the MoM estimators in the context of high-dimensional time series. We begin by investigating tail-robust mean estimation in Section 3.1, followed by an exploration of tail-robust autocovariance matrix estimation in Section 3.2.

## 3.1 Tail-robust Mean Estimation

Consider a univariate i.i.d. sample $\{X_i\}_{i=1}^n \subset \mathbb{R}$ of sub-Gaussian random variables with $\mathbb{E}(X_1) = \mu$ and $\text{var}(X_1) = \sigma^2$. As established by Devroye et al. (2016), the sample mean $\bar{X} = n^{-1} \sum_{i=1}^n X_i$ satisfies with probability at least $1 - \tau$ that

$$|\bar{X} - \mu| \leq \sqrt{\frac{2\sigma^2 \log(2/\tau)}{n}}. \tag{3.1}$$

In the multivariate case, let $\{X_i\}_{i=1}^n \subset \mathbb{R}^d$ be i.i.d. sub-Gaussian random vectors with mean vector $\boldsymbol{\mu} \in \mathbb{R}^d$. Without any additional assumptions on $\boldsymbol{\mu}$, using a union bound, the optimal coordinatewise error bound for the sample mean is of the order

$$|\bar{X} - \boldsymbol{\mu}|_\infty \lesssim \sqrt{\frac{\log(d/\tau)}{n}}. \tag{3.2}$$

In contrast, when $X_i$ is only assumed to have two finite moments, the achievable error rates of the sample mean are of

$$\sqrt{\frac{1}{n\tau}} \quad \text{and} \quad \sqrt{\frac{d}{n\tau}},$$

in 1-dimensional and $d$-dimensional settings, respectively (see e.g. Catoni, 2012).

Tail-robust estimators achieve the optimal error rates in (3.1) and (3.2), even when the underlying distributions have heavy tails. Although existing theoretical results primarily focus

on i.i.d. settings, we demonstrate that the MoM estimator retains its tail-robust properties in the presence of temporal dependence. The low-dimensional result below allows polynomial decay of the functional dependence measure, while the high-dimensional result imposes a geometric moment contraction condition to obtain exponential concentration after the median step. For comparison, previous studies such as Ke et al. (2019); Sun et al. (2020); Zhang (2021); Xu et al. (2023) have shown that Huber-type $M$-estimators are tail-robust under independence or stronger temporal dependence with exponential decay rates. In the remainder of this section, we introduce the MoM estimator and present its tail-robust properties under temporal dependence.

Let $\{X_i\}_{i=1}^n$ be a $d$-dimensional stationary time series, where $X_i = (X_{i,1}, \ldots, X_{i,d})^\intercal \in \mathbb{R}^d$, with mean vector $\boldsymbol{\mu} = (\mu_1, \ldots, \mu_d)^\intercal$. For each coordinate $j \in [d]$, the MoM estimator of $\mu_j$ is constructed as follows:

1. **Blocking:** Divide the sequence $\{X_{i,j}\}_{i=1}^n$ into $k$ consecutive blocks of equal size $m = \lfloor n/k \rfloor$, discarding any remaining observations outside these blocks.

2. **Block Means:** For each block $s \in [k]$, compute the block mean:

$$\bar{X}_j^s := \frac{1}{m} \sum_{i=1}^m X_{(s-1)m+i,j}.$$

3. **Median-of-Means:** Compute the MoM estimator $\bar{X}_j^{med}$, which is the median of the block means $\{\bar{X}_j^s\}_{s=1}^k$.

It is known that, under the i.i.d. settings, the MoM estimator $\bar{X}_j^{med}$ is tail-robust, achieving sharp error bounds requiring only that $\|X_{i,j}\|_2 < \infty$ (see e.g., Lugosi and Mendelson, 2019). Intuitively, computing block means reduces variance and symmetrizes the distribution, diminishing the difference between the median and the expectation of the block means. The blocking technique also reduces dependence between blocks. Consequently, the sample median of block means concentrates faster around the population median. However, non-asymptotic results for $\bar{X}_j^{med}$ under functional dependence have not been established prior to our work.

The following two theorems provide error bounds on

$$\hat{\boldsymbol{\mu}} := (\bar{X}_1^{\mathrm{med}}, \ldots, \bar{X}_d^{\mathrm{med}})^\intercal,$$

under two different conditions. These bounds are analogous to those in the i.i.d. settings (Lerasle and Oliveira, 2011). The first result uses polynomial decay of the functional dependence measure and is suited to moderate dimensions; the second replaces polynomial decay by GMC(2), which is enough to obtain the exponential-in-$k$ concentration required in the high-dimensional regime.

Theorem 3.1 requires only finite second moments and polynomial decay of the functional dependence measure in terms of the second moments, and yields a bound that is informative in the regime $d = o(n)$. Theorem 3.2 keeps the finite-second-moment requirement but strengthens the temporal-dependence condition to GMC(2), delivering a sharper bound that remains informative as long as $\log d = o(n)$; in contrast to the indicator-function based argument, a Lipschitz smoothing of the indicator (Lemma B.1 in the supplementary material) is used in the proof to remove any smoothness requirement on the marginal distribution of $X_{i,j}$.

Before stating the theorems, we need to define a quantity representing the block variance:

$$\sigma_{\text{block},j}^2 := \sup_{S \subseteq [n]} \frac{1}{\text{card}(S)} \mathbb{E}\big[\sum_{i \in S}(X_{i,j} - \mu_j)\big]^2,$$

which can be shown to be bounded using the maximal moment inequality under temporal dependence, i.e. Theorem 1 in Wu (2007). We also define $\sigma_{\text{block}}^2 := \max_{j \in [d]} \sigma_{\text{block},j}^2$. Recall that $\|X_{.,j}\|_{2,\nu}$ is the DAN(2) of $\{X_{i,j}\}_{i \in \mathbb{Z}}$, and $\|X_{.}\|_{2,\nu} = \max_{j \in [d]} \|X_{.,j}\|_{2,\nu}$ is the uniform version of $\|X_{.,j}\|_{2,\nu}$. Next, we introduce two assumptions for moments and coordinate-wise dependence measures, respectively.

**Assumption 1** (Marginal moments). $\omega_2^2 := \max_{j \in [d]} \omega_{2,j}^2 < \infty$, where $\omega_{2,j} := \|X_{i,j}\|_2 = (\mathbb{E}|X_{i,j}|^2)^{1/2}$ denotes the marginal $L_2$ norm of the $j$th coordinate.

**Assumption 2** (Temporal dependence). *(a) There exists some $\nu > 1$ such that $\|X_{.}\|_{2,\nu} < \infty$.*

*(b) There exists some $\rho \in (0,1)$ such that*

$$\|X_{.}\|_2 = \max_{j \in [d]} \sup_{m \geq 0} \rho^{-m} \sum_{i=m}^{\infty} \delta_{i,2,j} < \infty.$$

*Equivalently, the coordinate processes are uniformly GMC(2).*

Assumption 2(a) is a uniform DAN(2) condition with polynomially decaying cumulative functional dependence. Assumption 2(b) is stronger in the dependence direction, but not in the marginal moment direction: it keeps only second moments and replaces polynomial decay by geometric decay.

**Theorem 3.1** ( Low-dimension regime: $d = o(n)$). *Suppose Assumptions 1 and 2(a) hold. Choosing the number of blocks $k = \lceil \log n \rceil$ and assuming $3 \leq k \leq n/2$, we have with probability at least $1 - cd/n$*

$$|\hat{\boldsymbol{\mu}} - \boldsymbol{\mu}|_\infty \leq C \|X_{.}\|_{2,\nu} \sqrt{\frac{\{\log(n)\}^3}{n}},$$

*where $c, C > 0$ depend only on $\nu$. The probability statement is non-trivial precisely when $d = o(n)$.*

Next, we provide the second non-asymptotic result for the median-of-means estimator. This result is stronger than the one given in Theorem 3.1; it requires uniform GMC(2), but no moment higher than two. The proof, given in the supplementary material, relies on a Lipschitz approximation of the indicator function (see Lemma B.1 therein) whose functional dependence measure is controlled, up to a Lipschitz constant, by that of the block means alone—thereby eliminating the bounded-density assumption that an indicator-based argument would require on the marginal distribution of $X_{i,j}$.

**Theorem 3.2** ( High-dimension regime: $\log d = o(n)$). *Under Assumptions 1 and 2(b), choose the number of blocks $k = \lceil C_0 \log(n \vee d) \rceil$ and assume $3 \leq k \leq n/2$, where $C_0$ is sufficiently large and depends only on the GMC parameter $\rho$. we have with probability at least $1 - c(n \vee d)^{-3}$*

$$|\hat{\boldsymbol{\mu}} - \boldsymbol{\mu}|_\infty \leq C \|X_{.}\|_2 \sqrt{\frac{\log(n \vee d)}{n}},$$

where $c, C > 0$ *depend only on the GMC parameter* $\rho$, *and* $\|X_{\cdot}\|_2 = \max_{j \in [d]} \|X_{\cdot,j}\|_2$ *is the uniform coordinate-wise GMC(2) norm, not the Euclidean norm of* $\mathbf{X}_i$.

## 3.2 Tail-robust Autocovariance Matrix Estimation

In this subsection, we focus on the tail-robust MoM estimation for the autocovariance matrix, a key component in analyzing high-dimensional time series. This estimation has wide applications in fields such as finance, epidemiology, and genomics.

Let $\{\mathbf{X}_i\}_{i \in \mathbb{Z}}$ be an $\mathbb{R}^d$-valued stationary time series with mean $\boldsymbol{\mu} = (\mu_1, \mu_2, ..., \mu_d)^\intercal$ and lag-$l$ autocovariance matrix $\boldsymbol{\Sigma}_l := (\gamma_{l,(jk)})_{1 \leq j,k \leq d}$, where

$$\gamma_{l,(jk)} := \mathbb{E}[(X_{i,j} - \mu_j)(X_{i+l,k} - \mu_k)] = \mathbb{E}[X_{i-l,j}X_{i,k}] - \mu_j\mu_k. \tag{3.3}$$

Let $L \in \mathbb{N}^+$ be the maximum lag of the autocovariance matrices. Since $\boldsymbol{\Sigma}_l = \boldsymbol{\Sigma}_{-l}^\intercal$, we only consider $\boldsymbol{\Sigma}_l$ with $0 \leq l \leq L$. According to (3.3), estimating $\mathbb{E}[X_{i-l,j}X_{i,k}]$ and $\mu_j\mu_k$ can be treated separately with the same element-wise median-of-means estimator. Denote the lag-$l$ outer product as

$$\mathbf{H}_{i,l} = \mathbf{X}_{i-l}\mathbf{X}_i^\intercal, i \in \mathbb{Z},$$

and the lag-$l$ cross product for $(j,k)$−th coordinate is denoted as $H_{i,l,(jk)} = X_{i-l,j}X_{i,k}$. The element-wise MoM estimator for lag-$l$ autocovariance matrix is denoted by

$$\hat{\boldsymbol{\Sigma}}_l = (\hat{\gamma}_{l,(jk)})_{j,k \in [d]}.$$

Its entries are computed according to the following steps for each $j, k \in [d]$ as

1. Compute the MoM of $\{X_{i,j}\}_{i=1}^n$, denoted by $\bar{X}_j^{med}$;

2. Compute the MoM of the lag-$l$ cross products $\{H_{i,l,(jk)}\}_{i=l+1}^n$, denoted by $\bar{H}_{l,(jk)}^{med}$;

3. Compute $\hat{\gamma}_{l,(jk)} := \bar{H}_{l,(jk)}^{med} - \bar{X}_j^{med}\bar{X}_k^{med}$.

The non-asymptotic properties for $\hat{\boldsymbol{\Sigma}}_l$ are provided in the following theorem under mild assumptions.

**Assumption 3.** $\omega_4 := \max_{j \in [d]} \omega_{4,j} < \infty$, *where* $\omega_{4,j} := \|X_{i,j}\|_4 = (\mathbb{E}|X_{i,j}|^4)^{1/4}$ *denotes the marginal $L_4$ norm of the $j$th coordinate.*

**Assumption 4.** *There exists some* $\rho \in (0,1)$ *such that* $\|X_{\cdot}\|_4 = \max_{j \in [d]} \sup_{m \geq 0} \rho^{-m} \sum_{i=m}^{\infty} \delta_{i,4,j} < \infty$.

Assumptions 3 and 4 respectively impose fourth moments and uniform GMC(4). The bounded-density assumption is not needed; the proof again uses the Lipschitz smoothing argument from Lemma B.1 in the supplementary material, now applied to the lag-$l$ cross products.

**Theorem 3.3** ( Fixed $l$ and $\log d = o(n - l)$). *Under Assumptions 3 and 4, choose the number of blocks* $k = \lceil C_0 \log(n \vee d) \rceil$ *and assume* $3 \leq k \leq (n-l)/2$, *where $C_0$ is sufficiently large and*

*depends only on $\rho$. Then, with probability at least $1 - c(n \vee d)^{-3}$*

$$\|\hat{\boldsymbol{\Sigma}}_l - \boldsymbol{\Sigma}_l\|_{\max} \leq C\left\{\rho^{-l}\omega_4\|X_\cdot\|_4 + \|X_\cdot\|_4^2\sqrt{\frac{\log(n \vee d)}{n - l}}\right\}\sqrt{\frac{\log(n \vee d)}{n - l}},$$

*where $c, C > 0$ depend only on $\rho$.*

For fixed $l$ and fixed dependence and moment constants, when the block-size condition is feasible and $\log d = o(n - l)$, the error $\|\hat{\boldsymbol{\Sigma}}_l - \boldsymbol{\Sigma}_l\|_{\max}$ in Theorem 3.3 is of order $\sqrt{\log d/n}$, which is optimal in the minimax sense. In terms of consistency, the dimension $d$ is allowed to grow exponentially with $n$ as long as $(\log d)/n \to 0$.

## 4 Simulation study

In this section, we present three simulated examples to assess the empirical performance of the MoM estimator. In Section 4.1, we apply the MoM estimator to estimate the mean of a stationary time series contaminated by heavy-tailed noise. In Section 4.2, we extend the MoM estimator to estimate the mean vector of high-dimensional time series with heavy-tailed errors. Finally, in Section 4.3, we evaluate the MoM estimator's performance in estimating the autocovariance matrices of high-dimensional time series.

**Verification of the moment and temporal-dependence conditions.** We verify that each setting meets our theoretical conditions. The mean estimators (Theorems 3.1 and 3.2) require only a finite second moment (Assumption 1); the autocovariance estimator (Theorem 3.3) requires a finite fourth moment (Assumption 3). Since $\mathbb{E}|X|^r < \infty$ iff $r < \nu$ for a Student's $t_\nu$ and iff $r < \alpha$ for a Pareto of shape $\alpha$ (Log-Normal and Weibull have all moments finite), the autocovariance example uses a standardized $t_5$ and a Pareto(4.5), which have a finite fourth moment, while the mean examples may use the heavier $t_3$ and Pareto(3), for which a finite second moment already suffices. In particular, the Example 1 innovation is a Huber mixture $(1 - \varepsilon) N(0, 1) + \varepsilon F$ of $N(0, 1)$ and a centred (zero-mean) heavy-tailed noise $F$ (a Pareto with shape 3, a Log-Normal, or a Weibull); as all components have a finite second moment, the mixture satisfies Assumption 1. For the functional dependence measure, the VAR(1) processes in Examples 2 and 3 are geometrically decaying, hence GMC (Assumptions 2(b) and 4), as Theorems 3.2 and 3.3 require; Example 1 uses a linear process with polynomially decaying coefficients, satisfying only the weaker DAN(2) condition (Assumption 2(a)) and is thus covered by Theorem 3.1.

### 4.1 Example 1: mean estimation for contaminated time series

Data contamination, arising from errors and biases during data collection, processing, or analysis, can significantly affect statistical analyses and decision-making. Common sources of contamination include human errors, instrument malfunctions, and non-representative sampling. For example, Blanpied et al. (2017) highlighted substantial data contamination in clinical trials due to instrument errors, while Kruskal and Mosteller (1979) discussed the effects of non-representative sampling. In this study, we use a simulated example to evaluate the robustness of

the MoM estimator against data contamination and compare its performance with the sample mean estimator.

Consider the following linear process with polynomially decaying coefficients:

$$X_i = \sum_{l=0}^{\infty} a_l \, \epsilon_{i-l}, \qquad a_l = (1+l)^{-\beta}, \quad \beta = 2.5,$$

where $\{\epsilon_i\}$ is a white-noise innovation sequence with zero mean and unit variance, and in practice the summation is truncated at a large lag. Unlike an autoregressive model, whose functional dependence measure decays geometrically, this process has $\delta_{i,2} \asymp i^{-\beta}$, so its cumulative functional dependence decays only polynomially: it satisfies the DAN(2) condition (Assumption 2(a)) with index $\nu \leq \beta - 1 = 1.5 > 1$, but it is *not* GMC. This is precisely the short-range, polynomially dependent regime covered by Theorem 3.1 for the univariate mean, and is therefore a sharper test of our theory. We contaminate this data-generating process following the classical Huber $\varepsilon$-contamination model (Huber and Ronchetti, 2009): each innovation independently equals a centred heavy-tailed draw with probability $\varepsilon$ and a standard normal draw otherwise, that is, $\epsilon_i \sim (1 - \varepsilon) \, N(0,1) + \varepsilon \, F$, where $F$ is one of the centred heavy-tailed laws specified below. Because every such $F$ has a finite second moment, the contaminated innovation sequence retains a finite second moment, so the resulting linear process still satisfies Assumption 1 together with the DAN(2) condition of Assumption 2(a); the contaminated setting therefore remains entirely within the scope of Theorem 3.1. We set the sample sizes to $n = 50$ and $n = 100$, and apply contamination fractions $\varepsilon \in \{10\%, 25\%, 40\%\}$.

We consider three types of heavy-tailed distributions to generate the contaminated noise:

1. (Log-Normal). The contaminated noise follows a Log-Normal distribution with mean 1 and standard deviation 2.

2. (Pareto). The contaminated noise follows a Pareto distribution with shape parameter 3 and scale parameter 4.

3. (Weibull). The contaminated noise follows a Weibull distribution with shape parameter 0.3 and scale parameter 4.

All noises drawn from these distributions are centralized by subtracting their respective population means. The Log-Normal distribution is included because it is strongly skewed yet moment-regular (all moments finite); this shows that the robustness of the estimators is not an artefact of symmetric heavy tails but holds across skewness and tail regimes.

In each of the scenarios above, we simulate 200 replications to investigate the empirical distributions of the MoM estimator , computed with the theory-prescribed number of blocks $k = \lceil 2 \log(n) \rceil$ (Theorems 3.1 and 3.2), and the sample mean estimator. Figure 1 displays boxplots of the two estimators across the three heavy-tailed distributions for both $n = 50$ and $n = 100$. Each row corresponds to a distribution; the columns are grouped first by sample size ($n = 50$ and $n = 100$) and, within each size, by the three levels of data contamination (10%, 25%, and 40%). In all heavy-tailed cases, the MoM estimator exhibits consistent behavior across contamination levels, maintaining robustness even at higher contamination levels. In contrast,

the sample mean estimator shows considerably more variability, particularly at the 25% and 40% contamination levels, with a greater number of outliers and a larger interquartile range, especially in the Log-Normal and Weibull scenarios. These results highlight the robustness of the MoM estimator in handling data contamination, where it remains stable and accurate, while the sample mean estimator becomes increasingly affected by outliers and heavy-tailed noise.

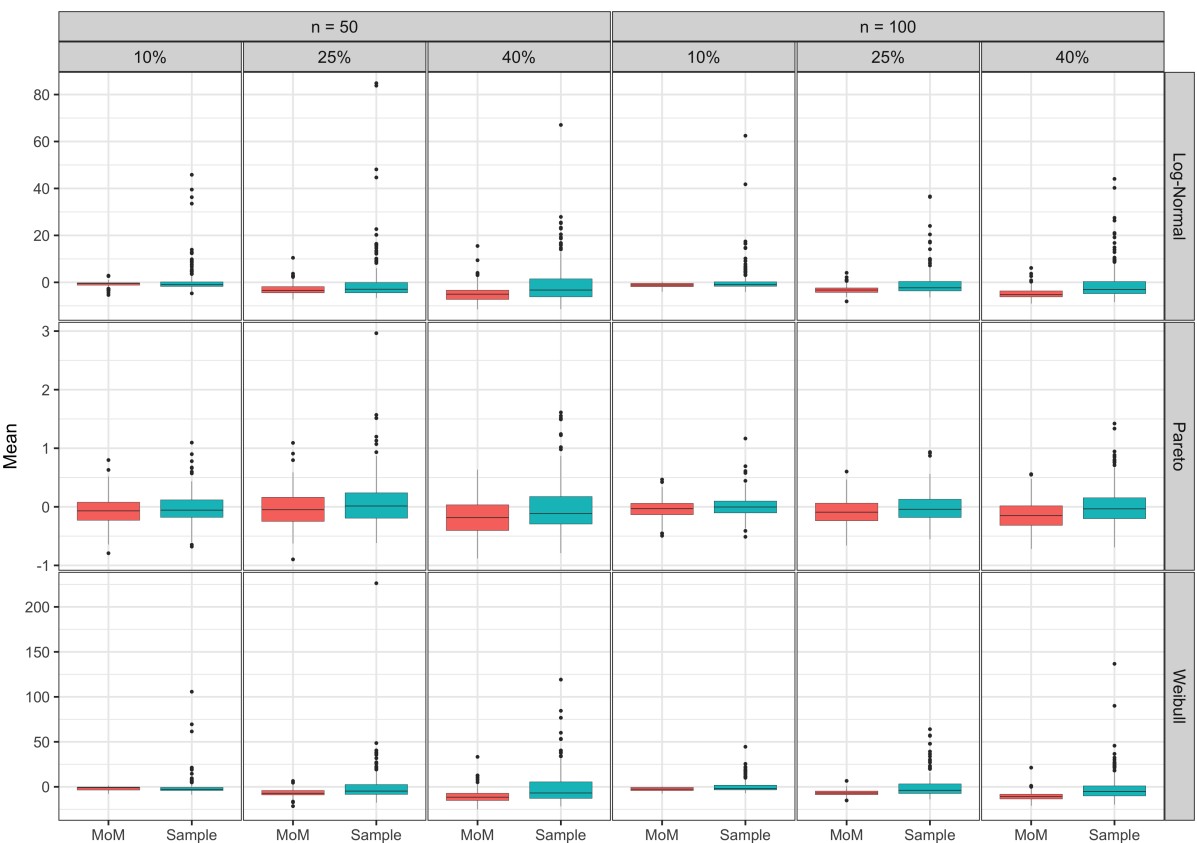

Figure 1: Example 1. boxplots of the MoM and sample-mean estimation errors over 200 replications. Rows correspond to the Log-Normal, Pareto, and Weibull contaminating distributions; columns are grouped by sample size ($n = 50$ and $n = 100$) and, within each size, by the contamination level (10%, 25%, and 40%). The MoM estimator stays tightly concentrated about the truth across all panels, whereas the sample mean develops a heavy spread and many extreme outliers as the contamination level rises, most visibly under the Log-Normal and Weibull laws.

## 4.2 Example 2: mean vector estimation for high-dimensional time series

The MoM estimator is also effective in high-dimensional settings. In this example, we validate the non-asymptotic properties of the MoM estimator for estimating the mean vector of high-dimensional time series and compare its performance to the classical sample mean and two state-of-the-art tail-robust estimators: the element-wise truncated estimator and the element-wise adaptive Huber estimator (Sun et al., 2020; Ke et al., 2019).

Consider generating an $\mathbb{R}^d$-valued time series from a vector autoregressive model defined as:

$$\mathbf{X}_i = \rho \mathbf{X}_{i-1} + \boldsymbol{\epsilon}_i,$$

where the scalar parameter $\rho = 0.5$. The error term $\boldsymbol{\epsilon}_i = (\epsilon_{i,1}, \ldots, \epsilon_{i,d})^\mathsf{T}$ is an i.i.d. sequence with $\mathbb{E}[\boldsymbol{\epsilon}_i] = 0$, and $\mathrm{var}(\boldsymbol{\epsilon}_i) = \mathbf{I}_d$, and $\mathbf{I}_d$ is the $d$ by $d$ identity matrix.

We consider the following four distributions of $\epsilon_{i,j}$:

1. (Normal). $\epsilon_{i,j}$ follows a standard Normal distribution.

2. (Pareto). $\epsilon_{i,j}$ follows a standardized Pareto distribution, i.e., $\epsilon_{i,j} = (12)^{-1/2}(Y_{i,j}-6)$ where $Y_{i,j}$'s are i.i.d. from a Pareto distribution with a shape parameter 3 and a scale parameter 4.

3. (Log-Normal). $\epsilon_{i,j}$ follows a standardized Log-normal distribution, i.e., $\epsilon_{i,j} = (e^8 - e^4)^{-1/2}[exp(Y_{i,j}) - exp(2)]$ where $Y_{i,j}$'s are i.i.d. from a Normal distribution $N(0,2)$.

4. (Student's t). $\epsilon_{i,j}$ follows a standardized Student's $t_3$ distribution, i.e., $\epsilon_{i,j} = 3^{-1/2}Y_{i,j}$ where $Y_{i,j}$'s are i.i.d. from a $t_3$ distribution.

For each combination of error distributions, we choose $(n, d)$ to be (50, 50), (50, 100), (50, 150), (100, 50), (100, 100), and (100, 150). We simulate 200 replications for each scenario. The MoM estimator uses the theory-prescribed number of blocks $k = \lceil 2\log(n \vee d) \rceil$ (Theorems 3.1 and 3.2). The sample mean vector estimator serves as the baseline for evaluating the performance of the MoM estimator and the other two tail-robust estimators. Specifically, we define the Relative Mean Error (RME) of the mean vector for the vector $l_\infty$-norm as follows:

$$\mathrm{RME} = \frac{\sum_{i=1}^{200} |\hat{\boldsymbol{\mu}}_{(i)} - \boldsymbol{\mu}|_\infty}{\sum_{i=1}^{200} |\tilde{\boldsymbol{\mu}}_{(i)} - \boldsymbol{\mu}|_\infty},$$

where $\hat{\boldsymbol{\mu}}_{(i)}$ is the mean vector estimator under consideration in the $i$-th replication, and $\tilde{\boldsymbol{\mu}}_{(i)}$ is the corresponding sample mean estimator. When RME $< 1$, the evaluated mean vector estimator outperforms the sample mean estimator.

Figure 2 shows the RMEs for the three competing estimators. In all cases, except for the Normal distribution, the MoM estimator and the other tail-robust estimators consistently outperform the sample mean across heavy-tailed distributions, particularly in the Pareto and Log-Normal scenarios. The MoM estimator performs comparably to the other two tail-robust estimators, confirming its effectiveness in high-dimensional settings with heavy-tailed noise.

**Remark 4.1.** *The MoM estimator is rate-optimal but not constant-optimal: it attains the minimax sub-Gaussian rate under only a finite second moment (mean) or a finite fourth moment (autocovariance), at the cost of a constant-factor efficiency loss under light or mildly heavy tails. For the high-dimensional* mean *(Figure 2) this is visible: under the Normal law the MoM estimator is worse than all competitors, and under heavy tails it beats the sample mean but is still worse than the truncated and Huber estimators, whose data-driven thresholds extract additional information. For the lag-l* autocovariance matrix *(Figure 3) the picture reverses: the entries are means of cross-products $X_{i,j}X_{i+l,k}$, whose effective tails are heavier than those of the coordinates, and the MoM estimator outperforms all competitors in most settings, including the Normal case.*

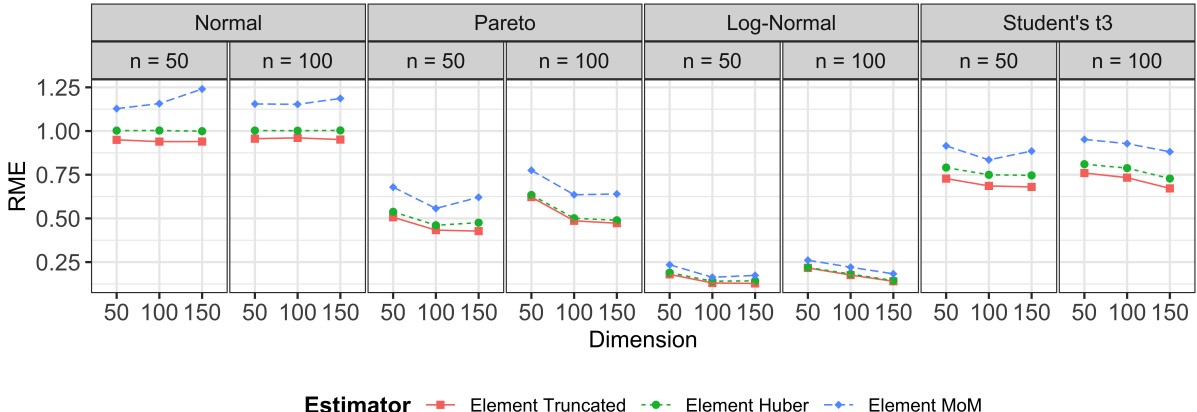

Figure 2: Example 2: RMEs of the three tail-robust mean vector estimators.

## 4.3 Example 3: autocovariance matrix estimation for high-dimensional time series

In this example, we validate the non-asymptotic properties of the MoM estimator for estimating the autocovariance matrices of high-dimensional time series. We compare the performance of the MoM autocovariance estimator to the classical sample autocovariance estimator, as well as two state-of-the-art tail-robust estimators: the element-wise truncated autocovariance estimator and the element-wise adaptive Huber autocovariance estimator (Xu et al., 2023).

Consider generating an $\mathbb{R}^d$-valued time series from a vector autoregressive model defined as:

$$\mathbf{X}_i = \rho \mathbf{X}_{i-1} + \mathbf{Z}_i,$$

where the scalar parameter $\rho = 0.5$. The sequence $\{\mathbf{Z}_i\}_{i \in \mathbb{Z}}$ is i.i.d., with $\mathbb{E}[\mathbf{Z}_i] = 0$ and $\mathrm{var}(\mathbf{Z}_i) = \boldsymbol{\Sigma} \in \mathbb{R}^{d \times d}$, where $\boldsymbol{\Sigma}$ is a deterministic matrix. Equivalently, we write $\mathbf{Z}_i = \boldsymbol{\Sigma}^{1/2} \boldsymbol{\epsilon}_i$, where $\boldsymbol{\epsilon}_i = (\epsilon_{i,1}, \ldots, \epsilon_{i,d})^\mathsf{T}$ are i.i.d. random vectors with $\mathbb{E}[\boldsymbol{\epsilon}_i] = 0$ and $\mathrm{var}(\boldsymbol{\epsilon}_i) = \mathbf{I}_d$. Since $\boldsymbol{\Sigma}$ is symmetric, the population lag-$l$ autocovariance matrix can be expressed as:

$$\boldsymbol{\Sigma}_l = (1 - \rho^2)^{-1} \rho^{|l|} \boldsymbol{\Sigma}.$$

We consider the following four distributions of $\epsilon_{i,j}$ :

1. (Normal). $\epsilon_{i,j}$ follows a standard Normal distribution.

2. (Pareto). $\epsilon_{i,j}$ follows a standardized Pareto distribution, i.e., $\epsilon_{i,j} = (36/245)^{-1/2}(Y_{i,j} - 9/7)$ where $Y_{i,j}$'s are i.i.d. from a Pareto distribution with a shape parameter 4.5 and a scale parameter 1.

3. (Log-Normal). $\epsilon_{i,j}$ follows a standardized Log-normal distribution, i.e., $\epsilon_{i,j} = (e^2 - e)^{-1/2}[\exp(Y_{i,j}) - \exp(1/2)]$ where $Y_{i,j}$'s are i.i.d. from a standard Normal distribution.

4. (Student's t). $\epsilon_{i,j}$ follows a standardized Student's $t_5$ distribution, i.e., $\epsilon_{i,j} = (5/3)^{-1/2}Y_{i,j}$ where $Y_{i,j}$'s are i.i.d. from a $t_5$ distribution.

Moreover, we consider the following three different structures for $\boldsymbol{\Sigma}$.

(a) Diagonal structure. $\mathbf{\Sigma} = \mathbf{I}_d$.

(b) Equal correlation structure. $\Sigma_{(ij)}$, the $(i, j)$-th element of $\mathbf{\Sigma}$, equals 1 if $i = j$ and equals 0.5 if $i \neq j$.

(c) Power decay structure. $\Sigma_{(ij)} = 0.5^{|i-j|}$.

For each combination of error distribution and $\mathbf{\Sigma}$ structure, we choose $(n, d)$ to be (50, 50), (50, 100), (50, 150), (100, 50), (100, 100), and (100, 150). We simulate 200 replications for each scenario. The MoM autocovariance estimator uses the theory-prescribed number of blocks $k = \lceil 2 \log(n \vee d) \rceil$ (Theorem 3.3). The sample autocovariance matrix estimator serves as the baseline for evaluating the performance of the MoM autocovariance estimator and the other two tail-robust autocovariance matrix estimators. Specifically, we define the Relative Estimation Error (REE) of the lag-$l$ autocovariance matrix for the max, spectral, or Frobenius norm as follows:

$$\text{REE}_l = \frac{\sum_{i=1}^{200} ||\hat{\mathbf{\Sigma}}_{l,(i)} - \mathbf{\Sigma}_l||_{\text{norm}}}{\sum_{i=1}^{200} ||\tilde{\mathbf{\Sigma}}_{l,(i)} - \mathbf{\Sigma}_l||_{\text{norm}}},$$

where $\hat{\mathbf{\Sigma}}_{l,(i)}$ is the lag-$l$ autocovariance matrix estimator under consideration in the $i$-th replication, and $\tilde{\mathbf{\Sigma}}_{l,(i)}$ is the corresponding sample autocovariance matrix estimator. When $\text{REE}_l < 1$ under a specific norm, the evaluated lag-$l$ autocovariance matrix estimator outperforms the sample autocovariance matrix estimator for that norm, and vice versa.

Figure 3 presents the REEs of the three competing lag-1 autocovariance matrix estimators. In this figure, red, green, and blue represent the three tail-robust autocovariance matrix estimators: 'Element Truncated', 'Element Adaptive Huber', and 'Element MoM'. Panels (a), (b), and (c) respectively display results for the diagonal, equal correlation, and power decay structures of $\mathbf{\Sigma}$. The results indicate that all three tail-robust estimators consistently outperform the sample autocovariance matrix estimator across all settings. In the Normal distribution scenario, the tail-robust estimators slightly outperform the sample autocovariance estimator, but their advantage becomes more pronounced in heavy-tailed settings. The MoM estimator performs comparably to the other tail-robust methods, with the added benefit of superior computational efficiency.

Table 1 reports the average computation time (in seconds) and standard deviation for the three estimation methods across the four heavy-tailed distributions for the $(n, d) = (50, 50)$ case. The MoM estimator consistently demonstrates a lower average computation time (around 0.10 seconds, roughly an order of magnitude faster than the other two methods) with small standard deviations, outperforming the other two methods in terms of efficiency.

# 5 Real data analysis

The S&P 500, established in 1957 and maintained by S&P Dow Jones Indices, is a widely followed stock market index consisting of 503 common stocks issued by 500 large-cap companies across 11 sectors of the U.S. economy. Covering approximately 80% of available market capitalization, it serves as a robust indicator of U.S. equities performance. Analyzing the historical performance of the S&P 500, particularly during periods of economic instability such as the 2007–2008 global financial crisis and the COVID-19 pandemic from 2019 to 2023, offers valuable insights into

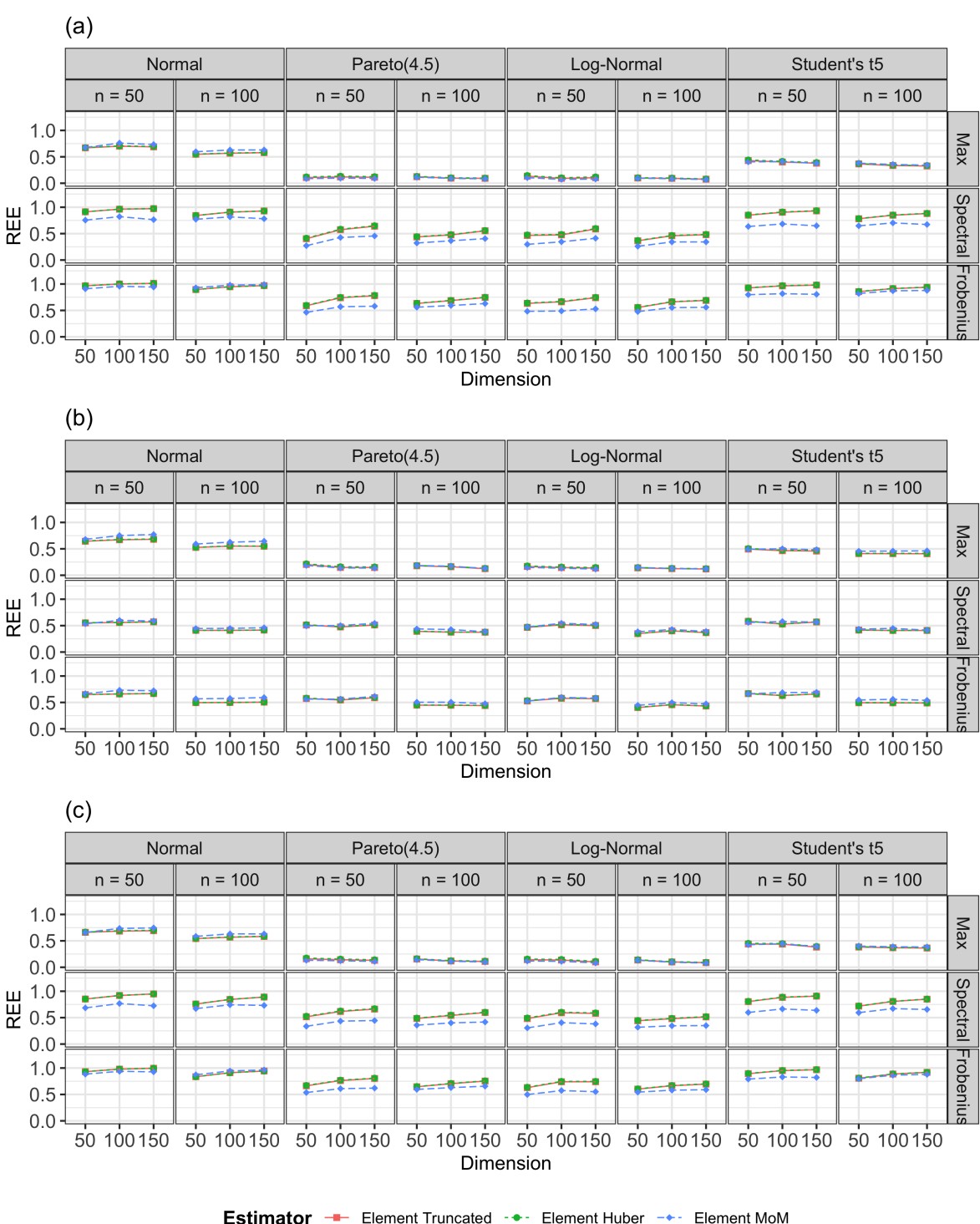

Figure 3: Example 3: REEs of the three tail-robust lag-1 autocovariance matrix estimators with $\mathbf{\Sigma}$ follows (a) the diagonal structure, (b) the equal correlation structure, and (c) the power decay structure.

|     |              | Element Truncated | Element Huber | Element MoM |
| --- | ------------ | ----------------- | ------------- | ----------- |
| (a) | Normal       | 0.83 (0.03)       | 0.88 (0.03)   | 0.10 (0.00) |
|     | Pareto       | 0.83 (0.03)       | 0.89 (0.04)   | 0.10 (0.01) |
|     | Log-Normal   | 0.83 (0.03)       | 0.88 (0.03)   | 0.10 (0.01) |
|     | Student's $t_5$ | 0.83 (0.03)    | 0.88 (0.03)   | 0.10 (0.00) |
| (b) | Normal       | 0.84 (0.04)       | 0.88 (0.03)   | 0.10 (0.01) |
|     | Pareto       | 0.83 (0.04)       | 0.88 (0.04)   | 0.10 (0.01) |
|     | Log-Normal   | 0.83 (0.03)       | 0.88 (0.04)   | 0.10 (0.01) |
|     | Student's $t_5$ | 0.84 (0.04)    | 0.88 (0.03)   | 0.10 (0.01) |
| (c) | Normal       | 0.84 (0.04)       | 0.88 (0.04)   | 0.10 (0.01) |
|     | Pareto       | 0.84 (0.03)       | 0.88 (0.04)   | 0.10 (0.01) |
|     | Log-Normal   | 0.83 (0.04)       | 0.88 (0.03)   | 0.10 (0.01) |
|     | Student's $t_5$ | 0.83 (0.03)    | 0.89 (0.04)   | 0.10 (0.01) |

Table 1: Example 3: A comparison of average computational time (in seconds) for $(n, d) = (50, 50)$ with (a) diagonal structure, (b) equal correlation structure, and (c) power decay structure. The numbers in parentheses are standard deviations.

market trends and investor sentiment. Applying the MoM estimation method to the S&P 500 log-return data ensures robustness and accurate trend representation. Significant growth phases, such as the recovery following the 2008 financial crisis, highlight the index's resilience, while periods of downturn emphasize market volatility and the importance of diversification. Thus, the S&P 500 serves not only as a measure of stock market performance but also as a window into the broader U.S. economy and the global financial landscape.

In this study, we analyze daily log-return data from 359 S&P 500 stocks with continuous records from January 2, 2004, to October 13, 2023. This high-dimensional time series spans major economic events over two decades, including the 2007–2008 global financial crisis and the COVID-19 pandemic. For each year in this interval, we compute the lag-$l$ autocovariance matrix using the element-wise MoM estimation method, as described in Section 3.2. We consider lags of $l = 1$ (daily) and $l = 5$ (weekly) to capture different time dependencies, and we vary the number of blocks (9, 11, 13, and 15) to assess their impact on the MoM estimation. The autocovariance matrices are analyzed using spectral, maximum, and Frobenius norms to explore trends over the years, providing insights into annual variations in market behavior.

Figure 4 presents the norms of autocovariance matrices from 2004 to 2023. Notably, a pronounced peak appears in 2008, aligning with the global financial crisis. During this period, market volatility surged as investors reacted to the collapse of financial institutions, credit tightening, and falling asset prices. This peak illustrates the sensitivity of the MoM method to drastic market shifts. Additionally, the results indicate that varying the number of blocks does not significantly alter the annual trends, affirming the robustness of the MoM estimator. A similar spike is observed in 2020, corresponding to the COVID-19 pandemic. The pandemic led to unprecedented market turbulence as global lockdowns, supply chain disruptions, and economic uncertainty caused sharp fluctuations in stock prices. This further demonstrates the method's effectiveness in capturing significant economic shocks.

Figure 5 shows heatmaps of the autocovariance matrices for 2007, 2008, and 2009, with lag $l = 1$ and 11 blocks. The color scale ranges from red (indicating higher positive values)

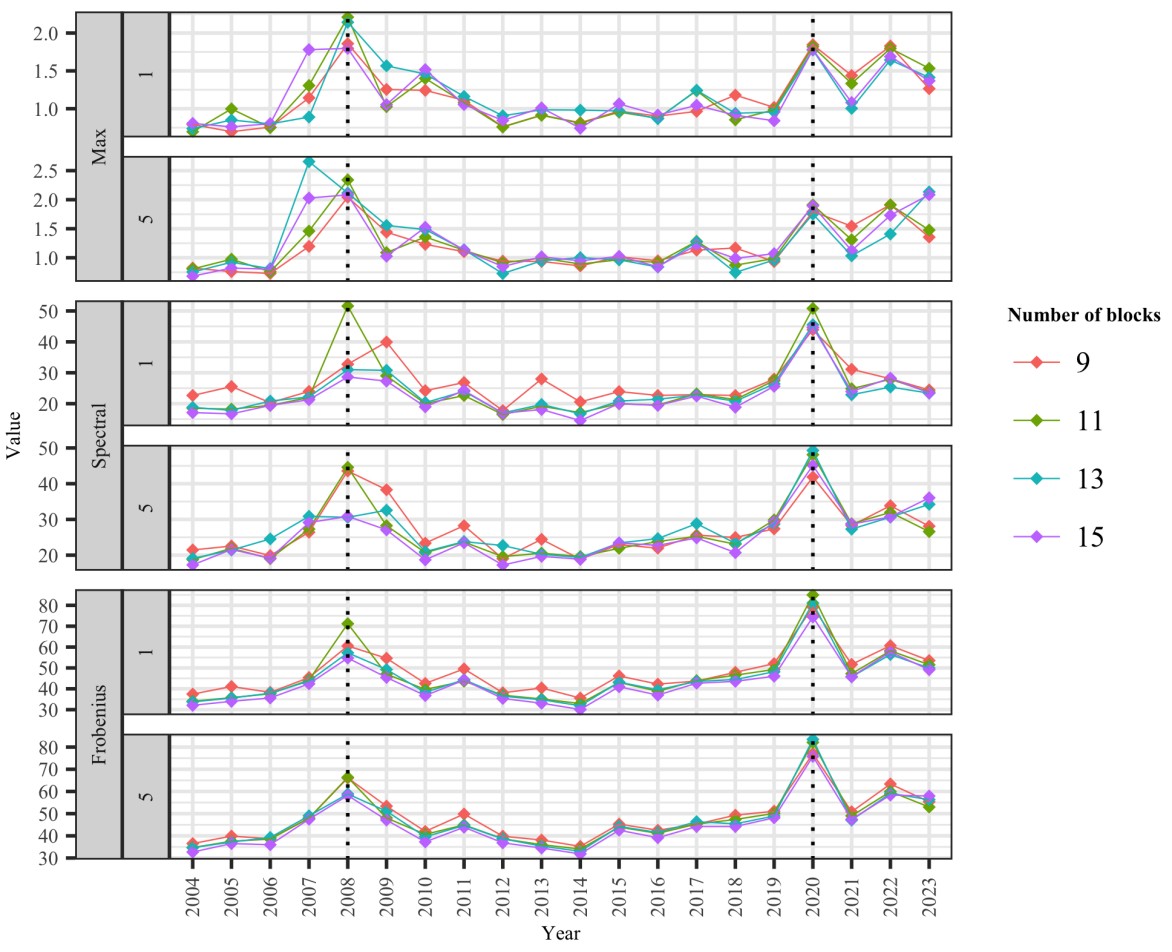

Figure 4: Annual trends in autocovariance matrix estimations based on median-of-means method for S&P 500 constituents (2004-2023) with lag $l = 1, 5$ and with respect to max, spectral and Frobenius norms. Four odd numbers of blocks are considered.

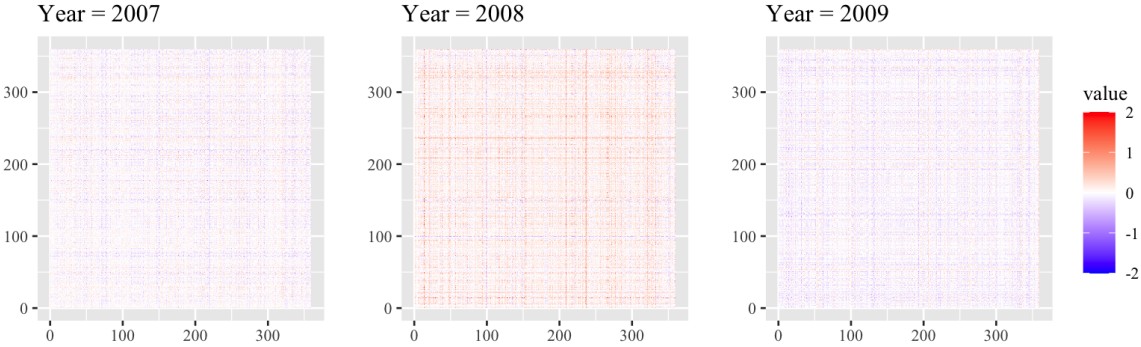

Figure 5: Lag-1 autocovariance matrix heatmap during financial crisis, year = 2007, 2008 and 2009.

to purple (indicating lower negative values), with white representing values near zero. The heatmap for 2007 reflects relative market stability, consistent with steady economic growth and low volatility. However, in 2008, the heatmap reveals extensive areas of high positive and negative autocovariances, corresponding to the severe market disruptions caused by the financial crisis. This year was marked by high volatility as major financial institutions collapsed and panic spread across markets. By 2009, the heatmap suggests partial stabilization, as financial rescue packages and monetary easing helped restore some market confidence and initiate economic recovery.

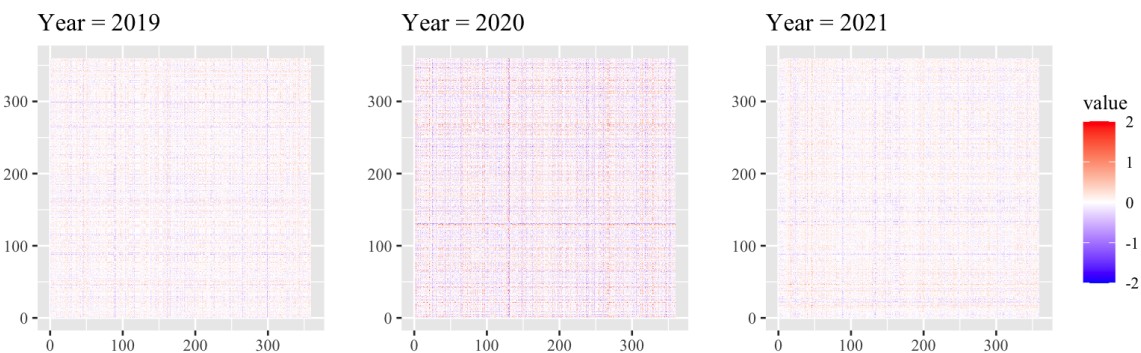

Figure 6: Lag-1 autocovariance matrix heatmap during COVID-19 pandemic, year = 2019, 2020, and 2021.

Similarly, Figure 6 presents heatmaps for the years 2019, 2020, and 2021. The 2019 heatmap shows relative market calm, reflecting a period of economic growth. However, in 2020, the heatmap reveals significant volatility due to the COVID-19 pandemic, which led to sharp market fluctuations as investors reacted to global lockdowns, supply chain disruptions, and widespread uncertainty. By 2021, the heatmap indicates reduced volatility, suggesting markets began to stabilize as vaccine rollouts and economic stimulus measures restored investors' confidence.

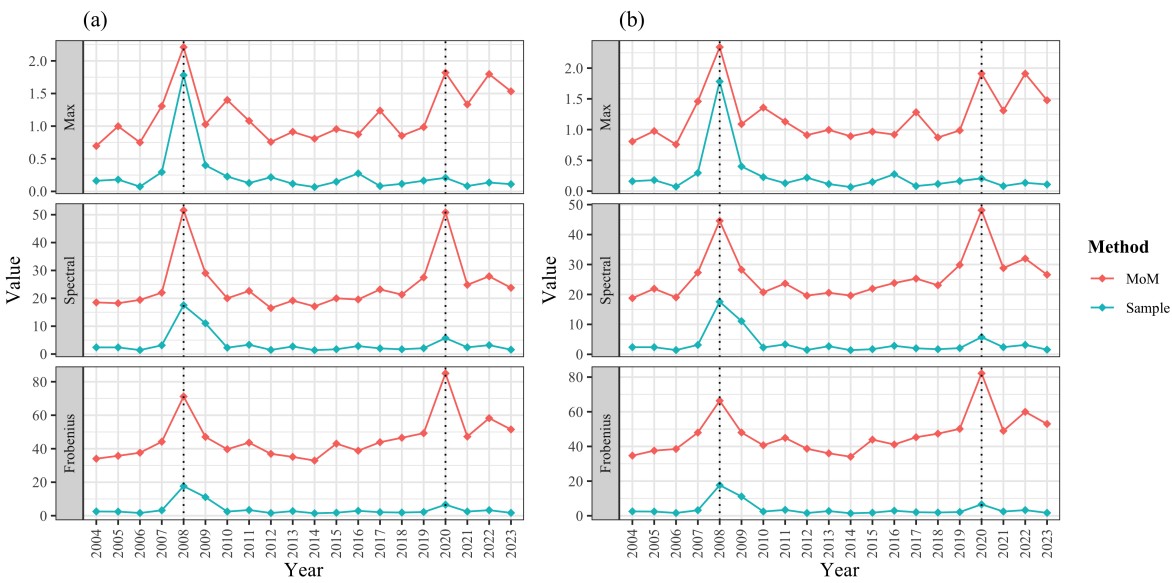

Figure 7: MoM estimator vs sample matrix with different lags: (a) l=1, (b) l=5.

Figure 7 compares the MoM estimation method to the classical sample autocovariance estimator by evaluating the max, spectral, and Frobenius norms of the autocovariance matrices across different lags. The MoM method exhibits greater variability in the autocovariance estimates compared to the classical method, reflecting its heightened sensitivity to market changes and significant events. The classical method tends to smooth over abrupt changes, potentially missing key inflection points in the data. The higher peaks observed in the MoM method during 2008 and 2020 underscore its ability to detect and respond to significant global events, such as the financial crisis and the COVID-19 pandemic. This sensitivity makes the MoM method a valuable tool for accurately modeling market risk and developing strategies that can adapt to sudden shifts, enhancing its utility in financial analysis and risk management.

## 6  Conclusions

In this study, we introduced a tail-robust autocovariance matrix estimator based on the MoM method for high-dimensional time series and provided a comprehensive analysis of its statistical properties. The performance of the estimator was examined from a non-asymptotic perspective, addressing challenges such as high-dimensionality, heavy-tailed distributions, and temporal dependence. We conducted extensive simulation experiments across various model settings to validate the theoretical results for both mean and autocovariance matrix estimation. The simulation results demonstrate that the proposed element-wise MoM estimator consistently outperforms the traditional sample autocovariance estimator and performs comparably to existing tail-robust autocovariance matrix estimators. Additionally, when applied to real-world data, the MoM estimator showed superior performance relative to the traditional approach, especially in capturing key market dynamics during periods of economic instability. These findings underscore the effectiveness of the MoM method in providing tail-robust, accurate estimates for high-dimensional time series data, making it a valuable tool for both theoretical analysis and practical applications in financial markets and beyond.

## Acknowledgment

The authors thank the reviewers for their constructive comments which led to significant improvement of this work. Li's research was supported by a NSF grant DMS 2514400 and two NIH grants R01GM163244 and R01 AI192205. Ke's research was supported by NSF grants 2210468, 2243044, 2324389, and 2514399, as well as an NIH grant R01-HL172291. Guerrier's research was funded by the Swiss National Science Foundation grants 233323, 232557, 10004089, and 216582.

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
