# Supplementary material for "Non–Asymptotic Analysis of Median-of-Means Estimation for High-Dimensional Time Series"

This supplementary material provides additional numerical results and detailed proofs supporting the theoretical results presented in this paper.

## A  Additional numerical results

This section is organized as follows. We first present supplementary results for the simulation and real-data studies of the main text: the lag-2 autocovariance estimators of Example 3 (Section A.1) and additional real-data heatmaps (Section A.2). We then report two further simulation examples: Example 4 (Section A.3), which verifies the convergence rate of the MoM mean estimator under genuine polynomial dependence, and Example 5 (Section A.4), which assesses the sensitivity of the MoM estimators to the number of blocks.

### A.1  Supplementary results for Example 3: lag-2 autocovariance

We report the lag-2 autocovariance results under the same theory-consistent setting as in the main text: the heavy-tailed innovations are the standardized Student's $t_5$ and Pareto (shape 4.5) laws, which possess a finite fourth moment as required by Theorem 3.3, and the MoM estimator uses the theory-prescribed number of blocks $k = \lceil 2\log(n \vee d) \rceil$. Figure A.1 summarizes the REEs of the three tail-robust lag-2 autocovariance estimators, showing that they consistently outperform the sample autocovariance matrix across all simulated settings. In the Normal distribution scenario, these tail-robust estimators slightly outperform the sample autocovariance matrix, and the outperformance becomes substantially more pronounced under the heavy-tailed Pareto and Student's $t_5$ laws, mirroring the lag-1 results in the main text. The MoM estimator remains comparable in accuracy to the other two tail-robust estimators while being roughly an order of magnitude faster (see Table 1), making it the preferred choice in high-dimensional settings.

### A.2  Supplementary real-data results

For the real-data analysis, we provide the median-of-means autocovariance matrix heatmap for $l = 5$ during the financial crisis and COVID-19 pandemic, as shown in Figures A.2 and A.3, respectively. These real-world applications, based on S&P 500 dataset, further highlight the median-of-means method's capability in detecting significant global market events.

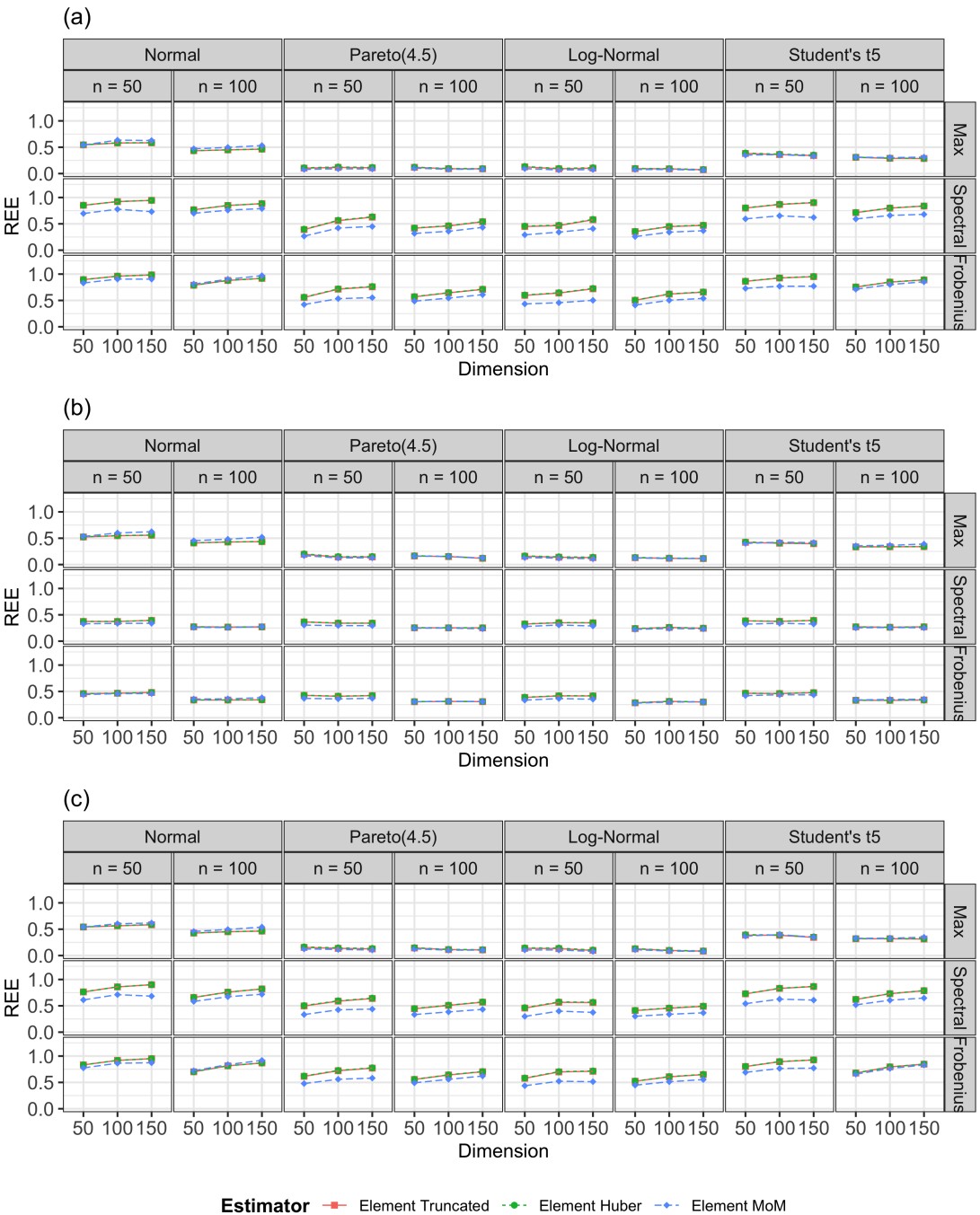

Figure A.1: Example 3: REEs of the three tail-robust lag-2 autocovariance matrix estimators with $\boldsymbol{\Sigma}$ follows (a) the diagonal structure, (b) the equal correlation structure, and (c) the power decay structure.

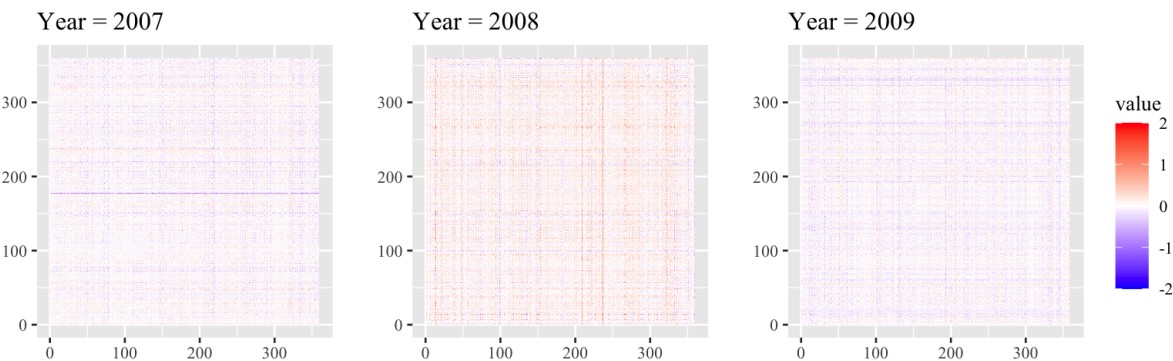

Figure A.2: Lag-5 autocovariance matrix heatmap during financial crisis, year = 2007, 2008 and 2009.

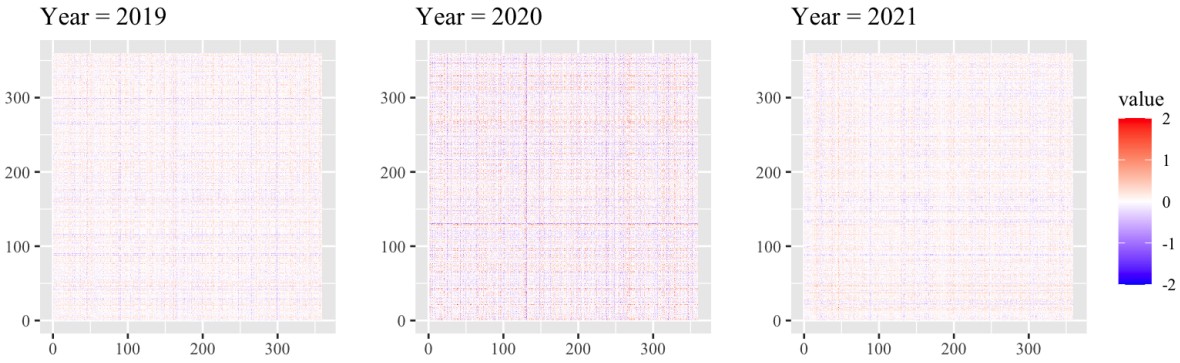

Figure A.3: Lag-5 autocovariance matrix heatmap during COVID-19 pandemic, year = 2019,2020, and 2021.

## A.3 Convergence under polynomial dependence (Example 4)

To complement Example 1, we further verify Theorem 3.1 directly by examining the convergence of the MoM mean estimator under genuine polynomial (DAN) temporal dependence. We generate the linear process $X_i = \sum_{l \geq 0}(1+l)^{-\beta}\varepsilon_{i-l}$ with true mean zero, for $\beta \in \{1.5, 2.5, 3.5\}$, dimension $d = 5$, sample sizes $n \in \{100, 200, 400, 800\}$, and the data-driven block count $k = \lceil 2\log n \rceil$; the innovations $\varepsilon_i$ are drawn from the Normal, Pareto, Student's $t_3$, and Log-Normal laws. Figure A.4 reports the mean absolute error of the MoM estimator on a logarithmic scale together with the theoretical rate.

Two conclusions emerge. First, for every $\beta$ and every innovation law the error decreases monotonically in $n$, confirming the consistency guaranteed by Theorem 3.1 under DAN(2) dependence (for example, at $\beta = 2.5$ the mean error under the Student's $t_3$ law decreases from 0.227 at $n = 100$ to 0.089 at $n = 800$, and the other laws behave analogously). Second, the error is inversely proportional to the value of beta, which matches the theory exactly: weaker temporal dependence yields a smaller error, and the $\beta = 1.5$ case—for which the dependence index $\nu < 1$, outside the scope of Theorem 3.1—is the slowest and serves as a designed stress boundary.

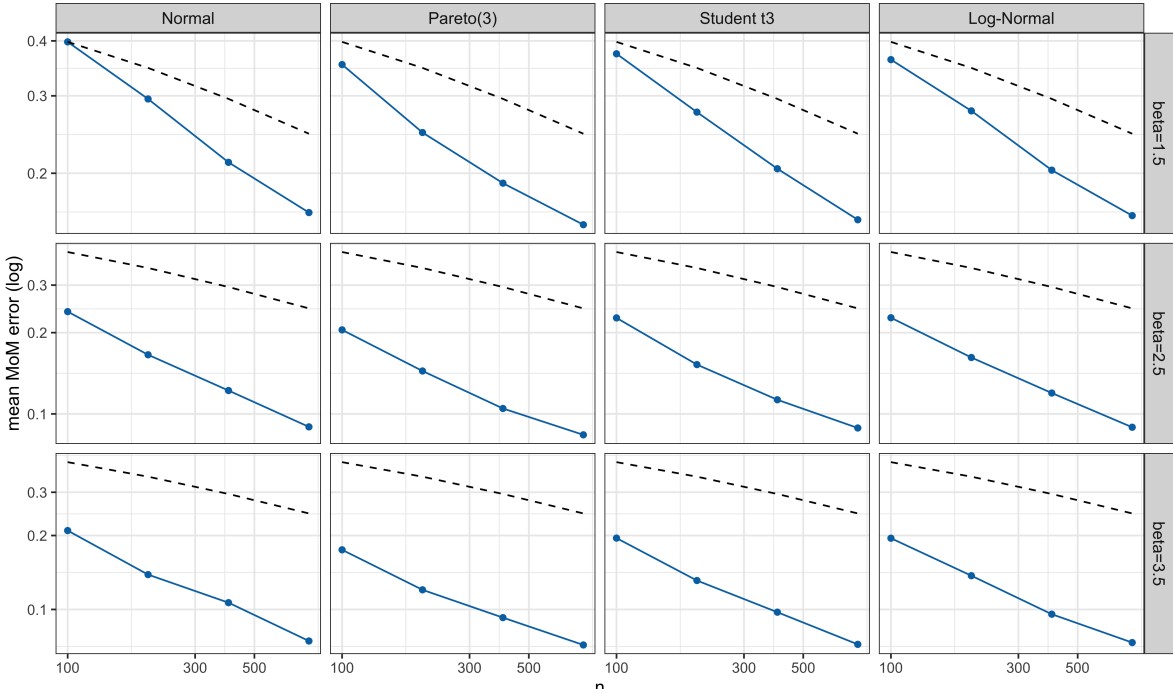

Figure A.4: Example 4: mean absolute error of the MoM mean estimator (logarithmic scale) versus $n$ under the polynomial-decay linear process, with the theoretical rate (dashed) overlaid, faceted by decay exponent $\beta$ (rows) and innovation law (columns).

## A.4 Sensitivity to the number of blocks and temporal dependence (Example 5)

The mechanism reuses Examples 2 and 3. For the mean we use the Example 2 VAR(1) process $\mathbf{X}_i = \rho\mathbf{X}_{i-1} + \boldsymbol{\epsilon}_i$ with $\mathrm{var}(\boldsymbol{\epsilon}_i) = \mathbf{I}_d$ and true mean zero, under the Normal, standardized $t_3$, and standardized Pareto (shape 3) laws (Example 2's finite-second-moment laws); for the autocovariance we use the same process with the diagonal structure $\boldsymbol{\Sigma} = \mathbf{I}_d$ of Example 3, so that $\boldsymbol{\Sigma}_1 = (1 - \rho^2)^{-1}\rho\,\mathbf{I}_d$, under the Normal, standardized $t_5$, and standardized Pareto (shape 4.5) laws (Example 3's finite-fourth-moment laws). The only change from Examples 2–3 is that the autoregressive coefficient $\rho$, fixed there at 0.5, is swept over $\{0.2, 0.5, 0.8, 0.9\}$ to vary the temporal dependence; we take $n \in \{100, 200\}$, $R = 200$ replications, $d = 50$ for the mean and 20 for the autocovariance, and sweep the block count $k$ from 1 (the sample estimator) to $k \approx n/2.5$. Figures A.5 and A.6 plot the resulting max-norm error against $k$ (vertical log scale) for the four dependence levels, with the *oracle k* the error-minimizing block count in each setting.

Two findings follow. First, the choice of $k$ is robust: the error-versus-$k$ curves are flat over a wide band, and the rule $k = \lceil 2\log(n \vee d)\rceil$ is within 13% (autocovariance) or 24% (mean) of the oracle once $\rho \geq 0.5$; how many blocks are needed is set by the tail, with $k = 1$ essentially the oracle under the light-tailed Normal mean but up to 3.4× worse under the heavy-tailed Pareto autocovariance, so blocking matters precisely when the tails are heavy. Second, the strength of temporal dependence acts on a different axis—the error *level* and an upper limit on $k$, not its optimal value. As $\rho$ grows the long-run variance grows (for an AR(1) the standard error scales like $(1 - \rho)^{-1}$), so the whole curve shifts up. Dependence also lowers the usable number of blocks, since each block must be long enough to decorrelate the series—the autocovariance oracle moves from $k \approx 66$ at $\rho = 0.2$ to $k \approx 12$ for $\rho \geq 0.5$, while the mean, which lacks temporal cross-products, is far less sensitive.

# B Technical Lemmas

For any $a \in \mathbb{R}$ and $\lambda > 0$, define the Lipschitz smoother

$$\psi_{a,\lambda}(u) = \begin{cases} 1, & u \leq a - \lambda, \\ (a - u)/\lambda, & a - \lambda < u \leq a, \\ 0, & u > a. \end{cases} \tag{B.1}$$

The function $\psi_{a,\lambda}$ replaces the indicator $\mathbb{1}_{\{u \leq a\}}$ in the proofs below, eliminating the need for a bounded-density assumption on the marginal distribution of $X_{i,j}$. The following lemma records its elementary properties.

**Lemma B.1.** *For any $a \in \mathbb{R}$ and $\lambda > 0$, the function $\psi_{a,\lambda} : \mathbb{R} \to [0, 1]$ defined in (B.1) satisfies:*

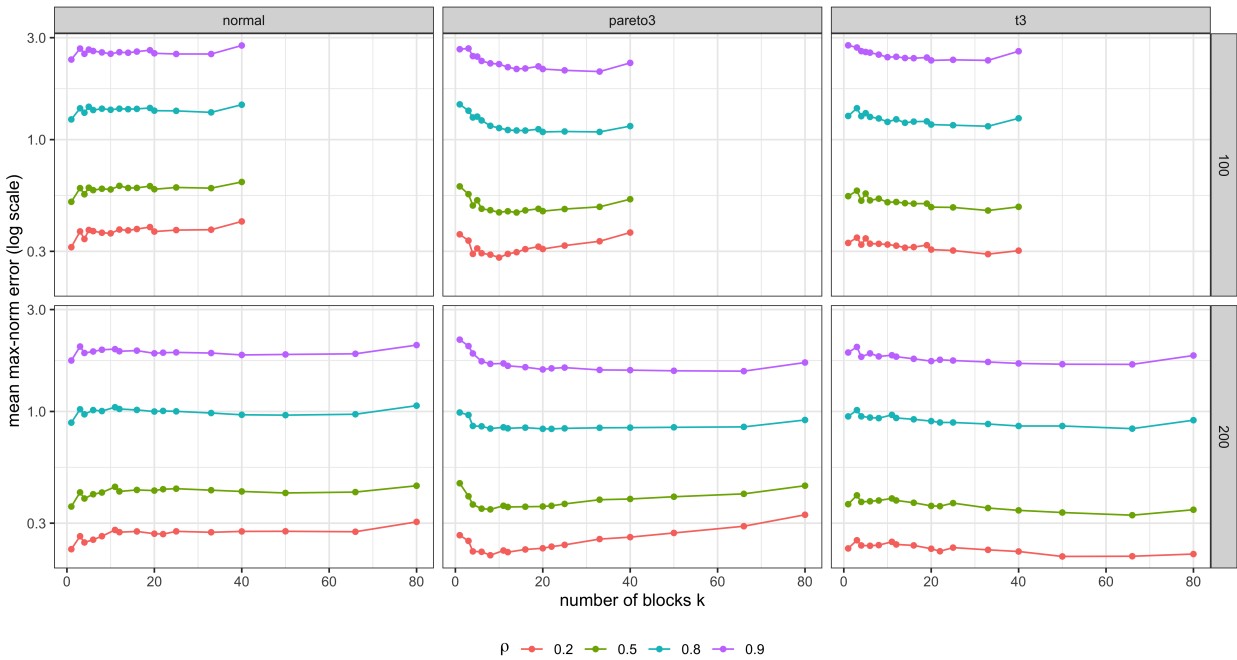

Figure A.5: Example 5: max-norm error of the MoM mean estimator versus the number of blocks $k$ (logarithmic vertical scale), for the four dependence strengths $\rho \in \{0.2, 0.5, 0.8, 0.9\}$ (colours), faceted by sample size $n$ (rows) and tail (columns). Each point is the average over the $R = 200$ replications of the max-norm error $\max_j |\bar{X}_j^{\text{med}} - \mu_j|$ at that number of blocks, with the points of a common $\rho$ joined by a line. The four $\rho$ curves are vertically separated, showing the error level rising with dependence, and $k = 1$ corresponds to the sample mean.

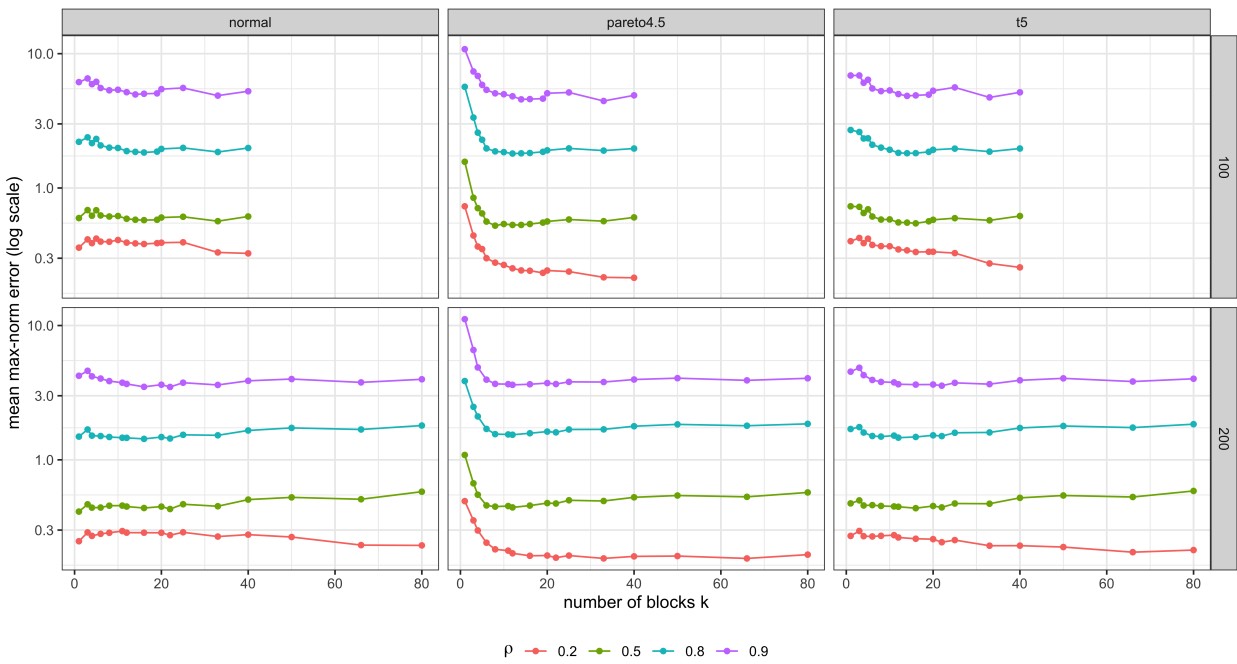

Figure A.6: Example 5: max-norm error of the MoM autocovariance estimator versus the number of blocks $k$ (logarithmic vertical scale), for the four dependence strengths $\rho \in \{0.2, 0.5, 0.8, 0.9\}$ (colours), faceted by sample size $n$ (rows) and tail (columns). Each point is the average over the $R = 200$ replications of the max-norm error $\|\hat{\boldsymbol{\Sigma}}_1 - \boldsymbol{\Sigma}_1\|_{\max}$ at that number of blocks, with the points of a common $\rho$ joined by a line, and $k = 1$ corresponds to the sample autocovariance.

(a) *(Sandwich)* $\mathbb{1}_{\{u \le a - \lambda\}} \le \psi_{a,\lambda}(u) \le \mathbb{1}_{\{u \le a\}}$ *for all* $u \in \mathbb{R}$.

(b) *(Lipschitz)* $|\psi_{a,\lambda}(u) - \psi_{a,\lambda}(v)| \le \lambda^{-1}|u - v|$ *for all* $u, v \in \mathbb{R}$.

*Consequently, for any pair of real-valued random variables* $Z, Z'$ *on the same probability space with* $\|Z - Z'\|_{q'} < \infty$ *for some* $q' \ge 1$,

$$\|\psi_{a,\lambda}(Z) - \psi_{a,\lambda}(Z')\|_{q'} \le \lambda^{-1}\|Z - Z'\|_{q'}. \tag{B.2}$$

*In particular, for any stationary sequence* $\{Z_i\}_{i \in \mathbb{Z}}$ *with coupled version* $\{Z'_i\}_{i \in \mathbb{Z}}$, *the functional dependence measure of* $\{\psi_{a,\lambda}(Z_i)\}_{i \in \mathbb{Z}}$ *is at most* $\lambda^{-1}$ *times that of* $\{Z_i\}_{i \in \mathbb{Z}}$.

*Proof of Lemma B.1. (a) Sandwich.* We verify the two inequalities by considering the three regions delineated by $a - \lambda$ and $a$.

- If $u \le a - \lambda$, then $\psi_{a,\lambda}(u) = 1$, while $\mathbb{1}_{\{u \le a-\lambda\}} = \mathbb{1}_{\{u \le a\}} = 1$. Both inequalities hold with equality.

- If $a - \lambda < u \le a$, then $\psi_{a,\lambda}(u) = (a-u)/\lambda$. Since $0 \le a - u < \lambda$, we have $\psi_{a,\lambda}(u) \in [0, 1)$. Moreover, $\mathbb{1}_{\{u \le a-\lambda\}} = 0$ and $\mathbb{1}_{\{u \le a\}} = 1$, so $0 \le \psi_{a,\lambda}(u) \le 1$ is exactly the required sandwich.

- If $u > a$, then $\psi_{a,\lambda}(u) = 0$ and $\mathbb{1}_{\{u \le a-\lambda\}} = \mathbb{1}_{\{u \le a\}} = 0$. All three quantities equal 0.

*(b) Lipschitz.* The function $\psi_{a,\lambda}$ is piecewise linear with pieces of slopes $0$, $-\lambda^{-1}$, and $0$ on the three regions above. It is continuous on $\mathbb{R}$: at the kink $u = a - \lambda$, both pieces give $\psi_{a,\lambda}(a - \lambda) = 1$; at the kink $u = a$, both pieces give $\psi_{a,\lambda}(a) = 0$. Hence $\psi_{a,\lambda}$ is absolutely continuous with derivative

$$\psi'_{a,\lambda}(u) = \begin{cases} 0, & u < a - \lambda, \\ -\lambda^{-1}, & a - \lambda < u < a, \\ 0, & u > a, \end{cases}$$

defined for all $u \in \mathbb{R} \setminus \{a - \lambda, a\}$ (a Lebesgue-null set), with $|\psi'_{a,\lambda}(u)| \le \lambda^{-1}$ wherever defined. By the fundamental theorem of calculus for absolutely continuous functions, for any $u \le v$,

$$|\psi_{a,\lambda}(u) - \psi_{a,\lambda}(v)| = \left| \int_u^v \psi'_{a,\lambda}(w)\, dw \right| \le \int_u^v |\psi'_{a,\lambda}(w)|\, dw \le \lambda^{-1}(v - u),$$

which gives (b) after taking $|u - v|$ in place of $v - u$ to allow either ordering of $u$ and $v$.

*Consequence* (B.2). Applying (b) pointwise to the random variables $Z, Z'$ yields

$$|\psi_{a,\lambda}(Z(\omega)) - \psi_{a,\lambda}(Z'(\omega))| \le \lambda^{-1}|Z(\omega) - Z'(\omega)|, \qquad \omega \in \Omega.$$

Since $\psi_{a,\lambda}$ is continuous (and hence Borel measurable), both sides are measurable. Raising to the power $q' \geq 1$ preserves the inequality, and taking expectations gives

$$\mathbb{E}|\psi_{a,\lambda}(Z) - \psi_{a,\lambda}(Z')|^{q'} \leq \lambda^{-q'} \, \mathbb{E}|Z - Z'|^{q'}.$$

Taking the $q'$-th root yields (B.2). The statement about functional dependence measures of stationary sequences is immediate by definition: $\delta_{i,q'}^{\psi(Z)} = \|\psi_{a,\lambda}(Z_i) - \psi_{a,\lambda}(Z_i')\|_{q'} \leq \lambda^{-1}\|Z_i - Z_i'\|_{q'} = \lambda^{-1}\delta_{i,q'}^Z$ for each $i$. $\qquad\square$

**Lemma B.2** (Theorem 1 in Wu (2007)). *Assume $\mathbb{E}X_i = 0$, $\|X_i\|_q < \infty$ and $\sum_{k=0}^{\infty} \delta_{k,q} < \infty$ for some $q \geq 2$, then, we have*

$$\|S_n^*\|_q \leq \frac{qB_q}{q-1} n^{1/2} \sum_{k=0}^{\infty} \delta_{k,q}, \tag{B.3}$$

*where $S_n^* = \max_{1 \leq r \leq n} |\sum_{i=1}^{r} X_i|$, $B_q = 18q^{3/2}(q-1)^{-1/2}$ if $q > 2$ and $B_q = 1$ if $q = 2$.*

The following lemma is the bounded-GMC specialization of the Bernstein inequality in Theorem 4 of Xu et al. (2024).

**Lemma B.3** (Bernstein inequality under GMC). *Let $\{Y_s\}_{s \in \mathbb{Z}}$ be a stationary, mean-zero causal process satisfying $|Y_s| \leq B$ almost surely. The innovations driving $\{Y_s\}$ may be vector-valued; $\delta_{\ell,2}^Y$ is computed by replacing the innovation at lag $\ell$ in this causal representation. Assume that, for some $\rho \in (0,1)$,*

$$\|Y_\cdot\|_{2,\rho} := \sup_{r \geq 0} \rho^{-r} \sum_{\ell=r}^{\infty} \delta_{\ell,2}^Y < \infty.$$

*Then there exist constants $C, c, c_0 > 0$, depending only on $\rho$, such that for any $k \geq 3$ and any $x \geq 1$,*

$$\mathbb{P}\left(k^{-1/2}\left|\sum_{s=1}^{k} Y_s\right| \geq C\{B + \|Y_\cdot\|_{2,\rho}\}x\right) \leq k\exp(-cx\sqrt{k}) + 2\exp(-cx^2). \tag{B.4}$$

*Consequently, for any $t \geq 1$ and $k \geq c_0\{t + \log k\}$,*

$$\mathbb{P}\left(\frac{1}{k}\sum_{s=1}^{k} Y_s \leq -C\{B + \|Y_\cdot\|_{2,\rho}\}\sqrt{\frac{t}{k}}\right) \leq Ce^{-t}. \tag{B.5}$$

*Proof of Lemma B.3.* Write $\Theta = \|Y_\cdot\|_{2,\rho}$ and $\widetilde{Y}_s = Y_s/(B + \Theta)$. Then $|\widetilde{Y}_s| \leq 1$ and

$$\sum_{\ell=r}^{\infty} \delta_{\ell,2}^{\widetilde{Y}} \leq \rho^r, \qquad r \geq 0.$$

Thus the cumulative $L_2$ functional dependence measure of $\{\widetilde{Y}_s\}$ decays exponentially with exponent $\gamma_1 = 1$. Since $\{\widetilde{Y}_s\}$ is bounded, it is the bounded-tail limiting case of the sub-Weibull setting in Theorem 4 of **?**; equivalently, the same display follows by their proof with no marginal truncation step. Applying that Bernstein inequality to $\{\widetilde{Y}_s\}$ gives (B.4) after rescaling. Taking $x = C'\sqrt{t}$ and choosing $C'$ and $c_0$ large enough so that $k \exp(-cC'\sqrt{tk}) \le e^{-t}$ whenever $k \ge c_0\{t + \log k\}$ gives (B.5). $\qquad\square$

# C  Proofs of Theorems

*Proof of Theorem 3.1.* Fix $j \in [d]$ and write $B_j = \|X_{.,j}\|_{2,\nu}$ and $B = \|X_.\|_{2,\nu}$. It suffices to prove the upper tail; the lower tail follows by applying the same argument to $\{-X_{i,j}\}$.

Let $m = \lfloor n/k \rfloor$. For each $s = 1, \ldots, k$, construct a coupled block mean $\bar{Z}_j^s$ separately as follows. Keep the innovations inside the $s$th block unchanged, and replace all innovations before that block by a copy independent of both the original innovations and the copies used for the other blocks. For $s = 1$, this means replacing the innovations prior to the first observed block by an independent copy. Then the block means $\{\bar{Z}_j^s\}_{s=1}^k$ are independent, each $\bar{Z}_j^s$ has the same marginal distribution as $\bar{X}_j^s$, and for every $s$,

$$\|\bar{X}_j^s - \bar{Z}_j^s\|_2 \le \frac{1}{m}\sum_{r=1}^m \sum_{u=r}^\infty \delta_{u,2,j} \le \frac{B_j}{m}\sum_{r=1}^m r^{-\nu} \le C_\nu B_j m^{-1}. \tag{C.1}$$

For $\varepsilon > 0$, let $A_s = \{|\bar{X}_j^s - \bar{Z}_j^s| \le \varepsilon\}$ and $A = \cap_{s=1}^k A_s$. By Markov's inequality and (C.1),

$$\mathbb{P}(A^c) \le C_\nu k B_j^2 m^{-2} \varepsilon^{-2}. \tag{C.2}$$

Choose $r_0 = (1 - \sqrt{1 - e^{-2}})/2$, so that $4r_0(1 - r_0) = e^{-2}$. By Chebyshev's inequality,

$$\mathbb{P}\{\bar{Z}_j^s - \mu_j > \|\bar{X}_j^s - \mu_j\|_2 r_0^{-1/2}\} = \mathbb{P}\{\bar{X}_j^s - \mu_j > \|\bar{X}_j^s - \mu_j\|_2 r_0^{-1/2}\} \le r_0.$$

Since the $\bar{Z}_j^s$ are independent,

$$\mathbb{P}\left(\#\left\{s : \bar{Z}_j^s - \mu_j > \|\bar{X}_j^s - \mu_j\|_2 r_0^{-1/2}\right\} \ge k/2\right) \le e^{-k}. \tag{C.3}$$

On $A$, if at least half of the original block means exceed $\mu_j + x$, then at least half of the coupled block means exceed $\mu_j + x - \varepsilon$. Combining this implication with (C.2) and (C.3), and taking $\varepsilon = C_\nu B_j k^{1/2} e^{k/2}/m$, gives

$$\bar{X}_j^{\text{med}} - \mu_j \le r_0^{-1/2}\|\bar{X}_j^s - \mu_j\|_2 + C_\nu B_j k^{1/2} e^{k/2} m^{-1}$$

with probability at least $1 - 2e^{-k}$.

By Lemma B.2 with $q = 2$,

$$\|\bar{X}_j^s - \mu_j\|_2 = m^{-1} \left\| \sum_{i=(s-1)m+1}^{sm} (X_{i,j} - \mu_j) \right\|_2 \leq C_\nu B_j m^{-1/2}.$$

Using $m \geq n/(2k)$ when $k \leq n/2$, the preceding display yields, for each coordinate,

$$|\bar{X}_j^{\mathrm{med}} - \mu_j| \leq C_\nu B \left\{ \sqrt{\frac{k}{n}} + \frac{k^{3/2} e^{k/2}}{n} \right\}$$

with probability at least $1 - 4e^{-k}$. With $k = \lceil \log n \rceil$, the right-hand side is bounded by $C_\nu B \sqrt{(\log n)^3/n}$, and $4e^{-k} \leq 4/n$. A union bound over $j \in [d]$ completes the proof. $\square$

*Proof of Theorem 3.2.* The argument adapts the usual median-of-means voting proof (Minsker, 2019; Lugosi and Mendelson, 2019). The independence-based concentration step is replaced by Lemma B.3, and the success probability of a single block is controlled only by Chebyshev's inequality. The key novelty is the use of the Lipschitz smoother $\psi_{a,\lambda}$ from Lemma B.1 in place of the indicator $\mathbb{1}_{\{u \leq a\}}$, which removes the bounded-density assumption on $X_{i,j}$. Throughout the proof, fix a coordinate $j \in [d]$.

**Reduction to two probability bounds.** Recall $\bar{X}_j^{\mathrm{med}} = \mathrm{med}(\bar{X}_j^1, \ldots, \bar{X}_j^k)$. Throughout the proof, $a > 0$ is treated as a free parameter; the smallest $a$ for which the high-probability bound $\bar{X}_j^{\mathrm{med}} \in [\mu_j - a, \mu_j + a]$ can be established will serve as the final upper bound on $|\bar{X}_j^{\mathrm{med}} - \mu_j|$. Equivalently, both events

$$\frac{1}{k} \sum_{s=1}^k \mathbb{1}_{\{\bar{X}_j^s - \mu_j \leq a\}} \geq \frac{1}{2} \quad \text{and} \quad \frac{1}{k} \sum_{s=1}^k \mathbb{1}_{\{\bar{X}_j^s - \mu_j \geq -a\}} \geq \frac{1}{2}$$

must hold with high probability. By symmetry it suffices to treat the first event; the second follows by replacing $X_{i,j}$ with $-X_{i,j}$.

**Smoothing the indicator.** By Lemma B.1(a), $\psi_{a,\lambda}(u) \leq \mathbb{1}_{\{u \leq a\}}$, so

$$\frac{1}{k} \sum_{s=1}^k \mathbb{1}_{\{\bar{X}_j^s - \mu_j \leq a\}} \geq W := \frac{1}{k} \sum_{s=1}^k \psi_{a,\lambda}(\bar{X}_j^s - \mu_j). \tag{C.4}$$

Let $\sigma_{s,j} := \|\bar{X}_j^s - \mu_j\|_2$. By stationarity, $\sigma_{s,j}$ and $\mathbb{E}\psi_{a,\lambda}(\bar{X}_j^s - \mu_j)$ do not depend on $s$; we retain the $s$-subscript for notational uniformity. Decompose

$$W = \underbrace{\frac{1}{k} \sum_{s=1}^k \left[ \psi_{a,\lambda}(\bar{X}_j^s - \mu_j) - \mathbb{E}\psi_{a,\lambda}(\bar{X}_j^s - \mu_j) \right]}_{=:I} + \mathbb{E}\psi_{a,\lambda}(\bar{X}_j^s - \mu_j). \tag{C.5}$$

We control $I$ via Lemma B.3. The deterministic lower bound for $\mathbb{E}\psi_{a,\lambda}(\bar{X}_j^s - \mu_j)$ is obtained from Chebyshev's inequality, so no moment above order two is needed.

**Expectation of the smoothed block success indicator.** Lemma B.1(a) gives

$$\mathbb{E}\psi_{a,\lambda}(\bar{X}_j^s - \mu_j) \geq \mathbb{P}(\bar{X}_j^s - \mu_j \leq a - \lambda) \geq 1 - \frac{\sigma_{s,j}^2}{(a-\lambda)^2}, \tag{C.6}$$

whenever $a > \lambda$, where the last inequality is Chebyshev's inequality applied to the upper tail.

**Term $I$ (concentration of the smoothed sum).** Let

$$D_s := \psi_{a,\lambda}(\bar{X}_j^s - \mu_j) - \mathbb{E}\psi_{a,\lambda}(\bar{X}_j^s - \mu_j).$$

We first verify that $\{D_s\}$ is GMC(2). View $\{D_s\}$ as a stationary causal process indexed by block time and driven by the i.i.d. block innovations $\mathcal{E}_s = (\epsilon_{(s-1)m+1}, \ldots, \epsilon_{sm})$. Let $\bar{X}_j^{s,(\ell)}$ be the block mean obtained after replacing the block innovation $\mathcal{E}_{s-\ell}$ by an independent copy, and define $D_s^{(\ell)} = \psi_{a,\lambda}(\bar{X}_j^{s,(\ell)} - \mu_j) - \mathbb{E}\psi_{a,\lambda}(\bar{X}_j^s - \mu_j)$. The Lipschitz property of $\psi_{a,\lambda}$ gives

$$\|D_s - D_s^{(\ell)}\|_2 \leq \lambda^{-1}\|\bar{X}_j^s - \bar{X}_j^{s,(\ell)}\|_2. \tag{C.7}$$

Under Assumption 2(b), $\sum_{u=r}^{\infty} \delta_{u,2,j} \leq \|X_{.,j}\|_2 \rho^r$ for all $r \geq 0$. Write

$$\delta_{\ell,2}^{\bar{X}} = \|\bar{X}_j^s - \bar{X}_j^{s,(\ell)}\|_2, \qquad \ell \geq 0.$$

For the current block, Lemma B.2 with $q = 2$ gives

$$\delta_{0,2}^{\bar{X}} \leq 2\|\bar{X}_j^s - \mu_j\|_2 \leq C_\rho m^{-1/2}\|X_{.,j}\|_2. \tag{C.8}$$

For $\ell \geq 1$, replacing $\mathcal{E}_{s-\ell}$ replaces $m$ original innovations. By the triangle inequality,

$$\delta_{\ell,2}^{\bar{X}} \leq m^{-1}\sum_{a=1}^{m}\sum_{b=1}^{m}\delta_{\ell m+a-b,2,j}, \qquad \ell \geq 1.$$

Since $\delta_{u,2,j} \leq \|X_{.,j}\|_2 \rho^u$, the geometric sums over $a$ and $b$ give, for every $\ell \geq 1$,

$$\delta_{\ell,2}^{\bar{X}} \leq C_\rho m^{-1}\|X_{.,j}\|_2 \rho^{(\ell-1)m}.$$

Consequently, for every $r \geq 1$,

$$\sum_{\ell=r}^{\infty}\delta_{\ell,2}^{\bar{X}} \leq C_\rho m^{-1}\|X_{.,j}\|_2 \rho^{(r-1)m},$$

and (C.8) handles $r = 0$. Hence

$$\sup_{r \geq 0} \rho^{-r} \sum_{\ell=r}^{\infty} \delta_{\ell,2}^{\bar{X}} \leq C_\rho m^{-1/2} \|X_{\cdot,j}\|_2,$$

because $m \geq 1$. Combining this display with (C.7),

$$\|D_\cdot\|_{2,\rho} := \sup_{r \geq 0} \rho^{-r} \sum_{\ell=r}^{\infty} \delta_{\ell,2}^{D} \leq C_\rho \lambda^{-1} m^{-1/2} \|X_{\cdot,j}\|_2. \tag{C.9}$$

Also $|D_s| \leq 1$ since $\psi_{a,\lambda} \in [0,1]$. Applying Lemma B.3 with $Y_s = D_s$ and using the block-size condition $k \geq c_0\{t + \log k\}$ gives, with probability at least $1 - Ce^{-t}$,

$$I = \frac{1}{k} \sum_{s=1}^{k} D_s \geq -C_1 \left(1 + \lambda^{-1} m^{-1/2} \|X_{\cdot,j}\|_2\right) \sqrt{\frac{t}{k}}. \tag{C.10}$$

**Choice of $\lambda$ and final rate.** If $\|X_{\cdot,j}\|_2 = 0$, then $X_{i,j} = \mu_j$ almost surely and the claim is trivial. Otherwise set

$$\lambda = m^{-1/2} \|X_{\cdot,j}\|_2, \qquad a = \lambda + 4\sigma_{s,j}. \tag{C.11}$$

Then (C.6) gives

$$\mathbb{E}\psi_{a,\lambda}(\bar{X}_j^s - \mu_j) \geq 15/16.$$

Combining this display with (C.4), (C.5), and (C.10), we have, with probability at least $1 - Ce^{-t}$,

$$\frac{1}{k} \sum_{s=1}^{k} \mathbb{1}_{\{\bar{X}_j^s - \mu_j \leq a\}} \geq \frac{15}{16} - C_1 \sqrt{\frac{t}{k}},$$

because $\lambda^{-1} m^{-1/2} \|X_{\cdot,j}\|_2 = 1$. Hence, if $k \geq C_2\{t + \log k\}$ with $C_2$ sufficiently large, the right-hand side is at least $3/4$. Therefore strictly more than half of the block means are at most $\mu_j + a$, and $\bar{X}_j^{\mathrm{med}} - \mu_j \leq a$.

It remains to bound $a$. By Lemma B.2 with $q = 2$ and the GMC(2) assumption,

$$\sigma_{s,j}^2 = \frac{1}{m^2} \left\| \sum_{i=1}^{m} (X_{i,j} - \mu_j) \right\|_2^2 \leq C_\rho m^{-1} \|X_{\cdot,j}\|_2^2.$$

Thus, using $m = \lfloor n/k \rfloor$,

$$a \leq C_\rho \|X_{\cdot,j}\|_2 m^{-1/2} \leq C_\rho \|X_{\cdot,j}\|_2 \sqrt{k/n},$$

for $k \leq n/2$. Applying the same argument to $\{-X_{i,j}\}$ gives the lower-tail bound. Consequently, for each fixed coordinate $j$,

$$|\bar{X}_j^{\mathrm{med}} - \mu_j| \leq C_\rho \|X_{\cdot,j}\|_2 \sqrt{k/n} \tag{C.12}$$

with probability at least $1 - Ce^{-t}$, provided $k \geq C_2\{t + \log k\}$ and $k \leq n/2$.

*Union bound.* Let $N = n \vee d$ and take $t = 4 \log N$. Since $k = \lceil C_0 \log N \rceil$, $C_0$ can be chosen large enough so that $k \geq C_2\{t + \log k\}$; also $k \leq C \log N$. Combining the symmetric upper- and lower-tail events, the per-coordinate bound holds with probability $\geq 1 - 2c_1 e^{-t}$. Applying a union bound over $j \in [d]$ inflates the failure probability by a factor of $d$:

$$\mathbb{P}\Big(\max_{j \in [d]} |\bar{X}_j^{\text{med}} - \mu_j| \text{ exceeds the bound}\Big) \leq 2c_1 d e^{-t} = 2c_1 dN^{-4} \leq 2c_1 N^{-3},$$

where the last inequality uses $d \leq n \vee d$. Finally, (C.12) and $k \leq C \log N$ give the stated rate. This concludes the proof. $\square$

*Proof of Theorem 3.3.* Let $N = n \vee d$, $n_l = n - l$, and

$$r_l = \sqrt{\frac{\log N}{n_l}}.$$

The condition $k = \lceil C_0 \log N \rceil \leq n_l/2$ ensures that the block sizes for both the original series and the lag-$l$ cross-product series are at least of order $n_l/k$.

First we verify the dependence condition for the cross products. For fixed $(j, k) \in [d]^2$, set

$$H_{i,l,(jk)} = X_{i-l,j}X_{i,k}, \qquad \eta_{l,(jk)} = \mathbb{E}H_{i,l,(jk)}.$$

Let $H'_{i,l,(jk)}$ be the coupled version obtained by replacing $\epsilon_0$ with an independent copy in the underlying process, and use the convention $\delta_{u,q,j} = 0$ for $u < 0$. By Hölder's inequality and stationarity,

$$\begin{aligned}
\|H_{i,l,(jk)} - H'_{i,l,(jk)}\|_2 &\leq \|X_{i-l,j} - X'_{i-l,j}\|_4\|X_{i,k}\|_4 + \|X'_{i-l,j}\|_4\|X_{i,k} - X'_{i,k}\|_4 \\
&\leq \omega_4\{\delta_{i-l,4,j} + \delta_{i,4,k}\}.
\end{aligned} \tag{C.13}$$

Therefore, under Assumption 4,

$$\sup_{m \geq 0} \rho^{-m} \sum_{i=m}^{\infty} \|H_{i,l,(jk)} - H'_{i,l,(jk)}\|_2 \leq C_\rho \rho^{-l} \omega_4 \|X_.\|_4. \tag{C.14}$$

Indeed, for the first term on the right side of (C.13), the convention $\delta_{u,4,j} = 0$ for $u < 0$ and Assumption 4 imply, for every $r \geq 0$,

$$\rho^{-r} \sum_{i=r}^{\infty} \delta_{i-l,4,j} \leq \begin{cases} \rho^{-r}\|X_.\|_4, & r \leq l, \\ \rho^{-r}\rho^{r-l}\|X_.\|_4, & r > l, \end{cases} \leq \rho^{-l}\|X_.\|_4.$$

The second term satisfies $\rho^{-r} \sum_{i=r}^{\infty} \delta_{i,4,k} \leq \|X_.\|_4 \leq \rho^{-l}\|X_.\|_4$. Combining the two bounds with (C.13) gives (C.14). Thus the $d^2$ coordinate processes $\{H_{i,l,(jk)} - \eta_{l,(jk)}\}$ are uniformly GMC(2), with uniform GMC norm bounded by the right side of (C.14).

Applying the coordinatewise argument in the proof of Theorem 3.2 to these $d^2$ cross-product coordinates, with sample size $n_l$ and $t = 5 \log N$, gives

$$\max_{j,k \in [d]} |\bar{H}_{l,(jk)}^{\mathrm{med}} - \eta_{l,(jk)}| \leq C_\rho \rho^{-l} \omega_4 \|X_\cdot\|_4 r_l \tag{C.15}$$

with probability at least $1 - CN^{-3}$. Indeed, the choice of $C_0$ makes $k \geq C\{t + \log k\}$, and the union bound over $d^2$ coordinates contributes at most $d^2 e^{-5 \log N} \leq N^{-3}$.

Next, by Assumption 4 and the monotonicity of $L_q$ norms,

$$\|X_\cdot\|_2 \leq \|X_\cdot\|_4.$$

Applying the same Theorem 3.2 argument to the $d$ original coordinates gives

$$\Delta_\mu := \max_{j \in [d]} |\bar{X}_j^{\mathrm{med}} - \mu_j| \leq C_\rho \|X_\cdot\|_4 \sqrt{\frac{\log N}{n}} \leq C_\rho \|X_\cdot\|_4 r_l \tag{C.16}$$

with probability at least $1 - CN^{-3}$.

On the event where (C.15) and (C.16) both hold, for every $(j,k)$,

$$\begin{aligned}
|\hat{\gamma}_{l,(jk)} - \gamma_{l,(jk)}| &\leq |\bar{H}_{l,(jk)}^{\mathrm{med}} - \eta_{l,(jk)}| + |\bar{X}_j^{\mathrm{med}} \bar{X}_k^{\mathrm{med}} - \mu_j \mu_k| \\
&\leq |\bar{H}_{l,(jk)}^{\mathrm{med}} - \eta_{l,(jk)}| + 2|\boldsymbol{\mu}|_\infty \Delta_\mu + \Delta_\mu^2.
\end{aligned}$$

Since $|\boldsymbol{\mu}|_\infty \leq \omega_4$, (C.15) and (C.16) imply

$$\|\hat{\boldsymbol{\Sigma}}_l - \boldsymbol{\Sigma}_l\|_{\max} \leq C_\rho \left\{ \rho^{-l} \omega_4 \|X_\cdot\|_4 + \|X_\cdot\|_4^2 r_l \right\} r_l.$$

A final union bound over the two high-probability events completes the proof. $\square$