# OpenReview forum: "Non-Asymptotic Analysis of Median-of-Means Estimation for High-Dimensional Time Series"
_SLADS/Section_A — Accepted by SLADS_Section_A_

### Review · Reviewer_LKyb · 2026-04-15

**Summary Of Contributions:**

This paper studies median-of-means estimators for high-dimensional mean vectors and autocovariance matrices under the functional representation of stationary time series. Specifically, the authors establish non-asymptotic error bounds in the presence of temporal dependence and heavy-tailed distributions, allowing for weaker dependence conditions (e.g., polynomial decay) than those commonly assumed in the literature. Numerical experiments and real applications show that the proposed method performs competitively with existing robust estimators.

**Audience:**

Yes

**Broader Impact Concerns:**

There are no ethical implications of the work.

**Claims And Evidence:**

Yes

**Requested Changes:**

1. The presentation of contributions remains somewhat abstract. Although the paper is built upon the functional representation in (2.1), it is not clear which commonly used time series models satisfy the stated moment and dependence conditions. Providing explicit examples (e.g., linear or nonlinear models) would significantly improve the clarity of the paper and help delineate the scope of the proposed framework.

2. The definitions of $\omega_{2,j}^2$, $\omega_{3,j}^3$, $\delta_{i,2}$, and $\delta_{i,3}$ in Assumptions 1 and 2 are not explicitly stated. I suggest that the authors include a remark to better illustrate Assumptions 1 and 2. Specifically, it would be beneficial to clarify the moment conditions imposed on $\mathbf{X}_i$ or $\epsilon_i$ (e.g., existence of second- or third-order moments) as well as the type of temporal dependence, such as whether the functional dependence measure satisfies a polynomial decay or a stronger exponential decay condition. Similar issues arise in Assumptions 4 and 5. In addition, I suggest that the authors include a brief remark comparing assumptions and convergence rates with those in existing seminal works on heavy-tailed time series, such as Wang and Tsay (2023).

3. The assumption on the number of blocks $k$ is specified as a function of $n$ and $d$ in the current theoretical development. However, it seems to me that the optimal choice of $k$ may also depend on the tail behavior of the data and the strength of temporal dependence. Intuitively, for light-tailed data, smaller values of $k$ (even $k=1$, in which case the MoM estimator reduces to the sample mean) may suffice, whereas heavier tails may favor larger $k$ to enhance robustness. A similar consideration may apply to stronger or weaker temporal dependence, since blocking also serves to mitigate dependence. In this sense, the choice of $k$ appears to involve a trade-off among tail behavior, temporal dependence, and sample size. I therefore encourage the authors to further explore whether alternative choices of $k$ could relax the required moment or dependence conditions, and, if possible, to include some discussion or simulation evidence on the sensitivity of the method to the choice of $k$.

4. In the simulation study, the log-normal distribution appears to represent a skewed distribution rather than a heavy-tailed distribution. However, the assumptions in the paper do not impose any symmetry condition on the density distributions. It would be helpful if the authors could clarify the motivation for including the log-normal distribution. In addition, I suggest that the authors explicitly verify whether the used simulation settings satisfy the imposed moment and temporal dependence conditions. From Figure 3, the MoM estimator appears to perform slightly worse than the competing methods in several settings. The authors should provide a clear explanation for this observation.

References
1. Wang, D., Tsay, R.S., 2023. Rate-optimal robust estimation of high-dimensional vector autoregressive models. The Annals of Statistics 51, 846 – 877.

**Strengths And Weaknesses:**

The paper contains substantial non-asymptotic theoretical contributions. However, the presentation of assumptions and notation could be further clarified. In addition, further theoretical investigation on the choice of the number of blocks $k$, as well as more comprehensive simulation studies, would help strengthen the paper.

---

> ### Author Response · Authors · 2026-05-23
> **Response to Reviewer LKyb, Part 1/3**
>
> **Response to Reviewer LKyb, Part 1/3**
>
> We sincerely thank the reviewer for the careful reading of our manuscript and for the constructive comments. We have revised the paper carefully, and the main revisions are highlighted in orange in the revised manuscript. Due to the character limit, we split our response into three consecutive comments.  **We have also included a PDF copy of the response letter in the supplementary material of the revised submission, where the equations are displayed more clearly.**
>
> ### Response to Requested Change 1
>
> We thank the reviewer for this helpful suggestion. We have revised the manuscript to make the contributions and scope of the proposed framework more explicit.
>
> First, we rewrote the contribution paragraph in the Introduction to clarify what is established in the paper. In particular, we now state that, to the best of our knowledge, non-asymptotic guarantees for median-of-means estimators of high-dimensional time series under the functional/physical dependence framework were previously unavailable. Our results fill this gap by establishing max-norm error bounds for the MoM estimator of the high-dimensional mean vector under both polynomially decaying dependence and GMC dependence, and for the lag-$l$ autocovariance matrix under GMC dependence. We also emphasize that the results require only weak moment conditions: finite second moments for mean estimation and finite fourth moments for autocovariance estimation.
>
> Second, following the reviewer’s suggestion, we added explicit examples of commonly used time series models that satisfy the stated dependence conditions. Specifically, in Section 2.2, we now state that stationary causal ARMA, (G)ARCH, threshold and bilinear autoregressions, and iterated random function/Markov models are GMC under standard parameter conditions, whereas linear processes with polynomially decaying coefficients satisfy the DAN condition but are generally not GMC. We also cite Wu (2005, 2011) for explicit calculations of the functional dependence measures.
>
> Finally, we added a paragraph at the beginning of Section 4 to verify the moment and temporal-dependence conditions for the simulation settings. This paragraph explains which examples are covered by Theorems 3.1--3.3 and why, thereby further delineating the practical scope of the proposed framework.
>
> ### Response to Requested Change 2
>
> We thank the reviewer for this helpful comment. We have revised the assumptions and related discussion to make the moment and dependence conditions more transparent.
>
> First, we made the relevant quantities explicit. In Section 2.2, we now define the coordinate-wise functional dependence measure as
>
> $$
> \delta_{i,q,j}:=\|X_{i,j}-X'_{i,j}\|_q,
> $$
>
> where $X'_i$ is the coupled version of $X_i$ obtained by replacing $\epsilon_0$ with an independent copy. In Assumption 1, we explicitly define
>
> $$
> \omega_{2,j}:=\|X_{i,j}\|_2=(\mathbb{E}|X_{i,j}|^2)^{1/2},
> \qquad
> \omega_2=\max_{j\in[d]}\omega_{2,j}.
> $$
>
> For autocovariance estimation, we similarly define
>
> $$
> \omega_{4,j}:=\|X_{i,j}\|_4=(\mathbb{E}|X_{i,j}|^4)^{1/4},
> \qquad
> \omega_4=\max_{j\in[d]}\omega_{4,j}.
> $$
>
> Second, we simplified and clarified the moment requirements in the revised theory. The revised manuscript no longer uses the former third-moment quantities $\omega_{3,j}$ and $\delta_{i,3}$, nor the sixth-moment quantities that appeared in the earlier version. The final results require only finite second moments for mean estimation and finite fourth moments for autocovariance estimation. We also revised the discussion following Assumptions 1--2 to clearly separate the marginal moment condition from the temporal-dependence condition: polynomially decaying functional dependence under the DAN condition for Theorem 3.1, and exponentially decaying dependence under the GMC condition for Theorems 3.2 and 3.3.

---

> ### Author Response · Authors · 2026-05-23
> **Response to Reviewer LKyb, Part 2/3**
>
> **Response to Reviewer LKyb, Part 2/3**
>
> ### Continued Response to Requested Change 2
>
> Following the reviewer’s suggestion, we also added a comparison with existing work on heavy-tailed time series, including Wang and Tsay (2023). In the Introduction, we now explain that Wang and Tsay (2023) establish rate-optimal robust estimation for structured high-dimensional VAR models via a constrained Yule--Walker step combined with a truncation-based autocovariance estimator, under a bounded $(2+2\epsilon)$-th moment condition and geometrically decaying $\alpha$- or $\beta$-mixing.
>
> We then clarify that our work is complementary. Their endpoint $\epsilon=1$ corresponds to the finite-fourth-moment condition under which our autocovariance estimator attains the optimal $\sqrt{\log d/n}$ rate, while their $\epsilon\in(0,1)$ regime covers sub-fourth-moment settings at slower rates. On the dependence side, our low-dimensional mean-estimation result permits genuinely polynomially decaying functional dependence, and our framework estimates the mean vector and autocovariance matrix directly rather than structured VAR parameters.
>
> ### Response to Requested Change 3
>
> We thank the reviewer for this insightful comment. We agree that the choice of the number of blocks $k$ reflects a trade-off among tail behavior, temporal dependence, and sample size. We have revised the manuscript to clarify the theoretical role of $k$ and added new empirical evidence on its sensitivity.
>
> First, on the theoretical side, the revised results use a principled logarithmic choice of the number of blocks. Specifically, Theorems 3.2 and 3.3 take
>
> $$
> k=\lceil C_0\log(n\vee d)\rceil,
> $$
>
> with the feasibility condition $3\le k\le n/2$ for mean estimation and $3\le k\le (n-l)/2$ for lag-$l$ autocovariance estimation. We clarified that the theory is developed under short-range temporal dependence, including summable polynomial functional dependence and GMC dependence. In this regime, the dependence strength affects the constants in the concentration bounds, while the leading error rate and the logarithmic order of $k$ remain unchanged.
>
> Second, we added discussion explaining how $k$ controls the robustness--efficiency trade-off. A smaller $k$ leads to longer blocks and greater efficiency under light-tailed data; in the extreme case $k=1$, the MoM estimator reduces to the sample mean. In contrast, a larger $k$ increases the robustness gained from the median step and is therefore more beneficial under heavier tails. We added this discussion in Remark 4.1, where we explain that the MoM estimator is rate-optimal but not necessarily constant-optimal, and that its finite-sample performance reflects the trade-off between robustness and efficiency.
>
> Third, following the reviewer’s suggestion, we added a new sensitivity analysis in Example 5 of the supplementary material. In this experiment, we vary $k$ for both mean and autocovariance estimation under Normal, Student’s $t$, and Pareto distributions, and under several levels of temporal dependence. The results support the reviewer’s intuition. The choice $k=1$ is competitive only under light tails, but can be substantially worse under heavy-tailed autocovariance estimation. In contrast, the default rule $k=\lceil 2\log(n\vee d)\rceil$ is consistently close to the error-minimizing choice across different tail behaviors, dependence strengths, and sample sizes. The error curves are also relatively flat around the optimum, indicating that the estimator is stable to moderate changes in $k$.
>
> Finally, regarding whether alternative choices of $k$ could relax the moment or dependence conditions, we added a brief discussion noting that the blocking and median steps are precisely what convert weak moment assumptions, such as finite second moments for mean estimation and finite fourth moments for autocovariance estimation, into exponential-type concentration bounds. A fully adaptive choice of $k$ that automatically adjusts to unknown tail behavior or dependence strength is an interesting direction for future work, but is beyond the scope of the present paper.

---

> > ### Comment · Reviewer_LKyb · 2026-06-09
> >
> > I appreciate the authors’ discussion on the role of $k$ in robustness, as well as the new sensitivity analysis reported in Example 5 of the supplementary material. However, the data-generating mechanism in this example is not fully clear. Please clarify whether the mean and autocovariance settings follow the same models as Examples 2 and 3, respectively. In addition, since the purpose of Example 5 is to study the choice of $k$, it would be useful to further examine how the strength of temporal dependence affects the choice of $k$. This would better clarify the trade-off among robustness, and temporal dependence.

---

> ### Author Response · Authors · 2026-05-23
> **Response to Reviewer LKyb, Part 3/3**
>
> **Response to Reviewer LKyb, Part 3/3**
>
> ### Response to Requested Change 4
>
> We thank the reviewer for these helpful comments. We have revised the manuscript to clarify the role of the Log-Normal distribution, verify the simulation settings against the assumptions, and explain the relative performance of the MoM estimator.
>
> First, regarding the Log-Normal distribution, we agree that it is included primarily to represent a highly skewed distribution, rather than a symmetric heavy-tailed distribution such as Student’s $t$. This choice is intentional, since our theory does not impose any symmetry condition on the marginal distributions. The Log-Normal distribution is skewed but moment-regular, with all moments finite. Including it, together with the Normal, Pareto, and Student’s $t$ distributions, allows us to assess the estimators across different regimes of skewness and tail behavior. We added this clarification in Example 1, where the Log-Normal distribution is introduced, and also in the verification discussion at the beginning of Section 4.
>
> Second, following the reviewer’s suggestion, we added a paragraph titled “Verification of the moment and temporal-dependence conditions” at the beginning of Section 4. This paragraph explicitly checks the simulation settings against the assumptions used in the theoretical results. In particular, we clarify that the mean-estimation examples require only finite second moments, while the autocovariance example requires finite fourth moments. Accordingly, the autocovariance simulations use standardized $t_5$ and Pareto distributions with finite fourth moments, whereas the mean-estimation examples may use heavier-tailed $t_3$ and Pareto distributions because finite second moments are sufficient. We also verify the temporal-dependence conditions: the VAR(1) processes in Examples 2 and 3 satisfy GMC-type dependence, while the linear process with polynomially decaying coefficients in Example 1 satisfies the weaker DAN condition.
>
> Third, we added an explanation for the cases where the MoM estimator is slightly less accurate than competing tail-robust methods. As discussed in the new Remark 4.1, the MoM estimator is rate-optimal but not necessarily constant-optimal. It achieves the minimax sub-Gaussian rate under weak moment conditions, but this robustness can come with a constant-factor efficiency loss under light-tailed or mildly heavy-tailed settings. For high-dimensional mean estimation, this explains why the MoM estimator can be less efficient than the adaptive Huber or truncated estimators, whose data-driven thresholds may extract additional information when the tails are not too severe.
>
> For autocovariance estimation, however, the entries are averages of cross-products $X_{i,j}X_{i+l,k}$, whose effective tails are heavier than those of the original coordinates. In this setting, the MoM estimator becomes more competitive and often outperforms the alternatives. This discussion appears in Remark 4.1, immediately after the simulation results in Example 2.
>
> We again thank the reviewer for the constructive comments, which helped us improve the clarity, theoretical presentation, and empirical discussion of the manuscript.

---

> > ### Comment · Reviewer_LKyb · 2026-06-09
> >
> > The authors have addressed my concern and I have no further comments on this point.

---

> > ### Comment · Reviewer_LKyb · 2026-06-09
> >
> > In the second paragraph of Section 4, fourth line from the bottom, “de- pendence” should be “dependence”. I have no further comments on this point.

---

> > > ### Author Response · Authors · 2026-06-09
> > > **Response to Reviewer LKyb**
> > >
> > > Thank you for pointing this out. We will correct this typo.

---

> ### Author Response · Authors · 2026-06-10
> **Response to Reviewer LKyb**
>
> We thank you for this constructive comment. We have revised Section A.4 of the supplementary material to make the data-generating mechanism in Example 5 explicit. The example reuses the VAR(1) settings from Examples 2 and 3: the mean-estimation setting follows Example 2, while the autocovariance-estimation setting follows Example 3 with $\mathbf\Sigma=\mathbf I_d$ and $\mathbf\Sigma_1=(1-\rho^2)^{-1}\rho,\mathbf I_d$. We have also clarified the innovation distributions used in each case.
>
> Following the reviewer’s suggestion, we have rerun the simulations by varying the temporal-dependence parameter ($\rho$) over {0.2,0.5,0.8,0.9}. The updated results are reported in the revised Figures A.5 and A.6. These results show how the oracle choice of the block number (k) changes with the strength of temporal dependence and better illustrate the trade-off between robustness and temporal dependence. In particular, while larger (k) may improve robustness under heavy-tailed distributions, very large (k) shortens the effective block length and can be less desirable when the temporal dependence is strong.

---

> > ### Comment · Reviewer_LKyb · 2026-06-11
> >
> > Thanks for your clarification. I have no further comments on this point.

---

> > ### Comment · Reviewer_LKyb · 2026-06-11
> >
> > Thanks for your clarification. I have no further comments on this point.

---

### Review · Reviewer_voTS · 2026-05-09

**Summary Of Contributions:**

This paper studies the Median-of-Means (MoM) estimator for mean vectors and autocovariance matrices in high-dimensional, heavy-tailed, and temporally dependent time series. The authors derive non-asymptotic error bounds under polynomial decay dependence, which is more realistic than the exponential decay assumptions commonly found in the literature. The theoretical results are novel and supported by extensive simulations and a real-data application to S\&P 500 returns. Overall, this paper is well-motivated and addresses an interesting problem.

**Audience:**

Yes

**Broader Impact Concerns:**

No significant ethical concerns are identified.

**Claims And Evidence:**

Yes

**Requested Changes:**

1. The paper promotes the Median-of-Means estimator as a tail-robust method
requiring only mild moment conditions. For auto-covariance
estimation (Section~3.2),  Assumption 4 requires $\max_{j\in[d]} \|X_{i,j}\|_6 < \infty$, i.e., finite sixth moments.
This is a substantially stronger condition.  Moreover,
 $t_3$ distributions used in the simulation study may violate Assumptions 1(b) and 4, which are required for Theorems 3.2 and 3.3. The authors should either replace it with ones that satisfy the stated moment conditions or relax the theoretical assumptions accordingly.



2. The authors claim a major theoretical advance in handling polynomially decaying temporal dependence, yet all simulations use only an AR(1) model—which exhibits exponential, not polynomial, decay. No experiment features a process with true polynomial decay. Therefore, the simulations fail to validate the method under the dependence structure that motivates the theory. I recommend adding at least one simulation with genuine polynomial decay to substantiate the theoretical contribution.


3. Dividing the sequence into blocks introduces a nontrivial bias–variance tradeoff that is not fully explored. Shorter blocks may reduce the impact of contamination within any single block, but they also reduce the accuracy of block-wise mean/autocovariance matrix estimation. Longer blocks improve local estimation precision but weaken the robustness motivation. This tradeoff seems central to the proposed method, yet I did not find a sufficiently sharp treatment of how the block length should be selected in practice, nor a persuasive sensitivity analysis showing that the method is stable to this choice.


4.Theorems 3.2 and 3.3 require the number of blocks $k$ to satisfy theoretical ranges such as $\log(n) \lesssim k \lesssim \sqrt{n\log(n)}$ or $\log(n\vee d) \lesssim k \lesssim \sqrt{n\log(n\vee d)}$. However, the paper provides no data-driven rule for selecting $k$ in practice. The authors should provide a practical guideline or sensitivity analysis for choosing $k$.


5. Page 7, Assumption 1, the quantities $\omega_{2,j}$ and $\omega_{3,j}$ are used without definition. Please define them explicitly.


6.Page 4, Section 2.2: "wo commonly considered decay rates'' contains a typo ("wo").


7. Page 14, Case 3,  the  distribution is specified as a standardized Student's $t_3$. However, in Table 1 and Figure 4, the same distribution is labeled as Student's $t_4$. Please check it.

**Strengths And Weaknesses:**

Strengths：
1. Theoretical novelty: First non-asymptotic analysis of MoM for high-dimensional time series under polynomial decay dependence, relaxing the exponential decay assumption in prior work.
2. Solid empirical validation: Extensive simulations and a real S&P 500 application demonstrate robustness and practical value.

Weaknesses：
1. Theory-simulation mismatch: Theorems 3.2--3.3 require finite sixth moments, but the simulations use a Student's $t_3$ distribution. The authors should either replace the distribution or relax the assumptions.

 2.Lack of guidance on choosing $k$: The number of blocks $k$ has theoretical ranges (e.g., $\log(n\vee d) \lesssim k \lesssim \sqrt{n\log(n\vee d)}$), but no practical data-driven selection rule is provided.

3. Several typographical errors

---

> ### Author Response · Authors · 2026-05-23
> **Response to Reviewer voTS, Part 1/3**
>
> **Response to Reviewer voTS, Part 1/3**
>
> We are grateful for your thoughtful evaluation and detailed suggestions, which helped us improve the paper. We have carefully considered your comments and revised the manuscript accordingly. The main revisions are highlighted in orange. Due to the character limit, we split our response into three comments. **We have also included a PDF copy of the response letter in the supplementary material of the revised submission, where the equations are displayed more clearly.**
>
> ### Response to Requested Change 1
>
> We thank the reviewer for this helpful comment. We agree that the previous version imposed stronger-than-necessary moment assumptions, especially for autocovariance estimation, and that some simulation settings needed to be better aligned with the stated theory. We have substantially revised both the theoretical assumptions and the simulation discussion.
>
> First, for high-dimensional mean estimation, the revised Theorem 3.2 now requires only finite second moments together with a GMC(2) dependence condition. The previous stronger moment and bounded-density requirements have been removed. Technically, this improvement is obtained by replacing the discontinuous indicator function in the median-of-means voting argument with a Lipschitz-smoothed approximation. This allows us to control the functional dependence measure directly and eliminates the need for additional smoothness assumptions on the marginal density. The revised assumption and theorem appear on page 8, in Assumptions 1--2 and Theorem 3.2.
>
> Second, for autocovariance estimation, we have removed the previous finite sixth-moment requirement. The revised Theorem 3.3 now assumes only finite fourth moments and GMC(4), which is the natural condition needed to control the second moment of the lagged cross-products \(X_{i-l,j}X_{i,k}\). The autocovariance assumptions and proof have been updated accordingly to verify that the cross-product process satisfies the required GMC(2)-type condition under finite-fourth-moment and GMC(4) assumptions. These revisions appear on pages 9--10, in Assumptions 3--4 and Theorem 3.3.
>
> Third, we revised the simulation settings and added an explicit verification paragraph at the beginning of Section 4. For mean estimation, the revised theory requires only a finite second moment, so the standardized \(t_3\) distribution and the Pareto distribution with shape parameter 3 are admissible. Indeed, the \(t_3\) distribution has finite variance but infinite third moment, making it a useful example for illustrating the weakened moment requirement. For autocovariance estimation, Theorem 3.3 requires finite fourth moments. We therefore use a standardized \(t_5\) distribution and a Pareto distribution with shape parameter 4.5, both of which have finite fourth moments while remaining genuinely heavy-tailed. This verification appears on page 10, in the paragraph titled “Verification of the moment and temporal-dependence conditions,” and the revised autocovariance simulation distributions are stated on page 14 in Example 3.
>
> ### Response to Requested Change 2
>
> We thank the reviewer for this helpful suggestion. We agree that the original simulation design did not fully reflect the polynomially decaying dependence structure emphasized in part of the theory, since the AR(1) model has geometrically decaying dependence.
>
> In the revised manuscript, we clarified the dependence regimes covered by our theoretical results. Specifically, Theorem 3.1 allows polynomially decaying functional dependence under the DAN condition, while Theorems 3.2 and 3.3 are stated under GMC-type dependence conditions for the high-dimensional mean and autocovariance estimators. This clarification appears on pages 7--9, in the discussion preceding Theorems 3.1--3.3.
>
> To better align the numerical study with the theory, we revised Example 1 so that it now uses a genuinely polynomially dependent linear process,
>
> $$
> X_i=\sum_{l=0}^{\infty} a_l\epsilon_{i-l},
> \qquad
> a_l=(1+l)^{-\beta}, \quad \beta=2.5.
> $$
>
> For this process, the functional dependence measure satisfies \(\delta_{i,2}\asymp i^{-\beta}\), and the cumulative dependence measure decays polynomially. Hence the process satisfies the DAN(2) condition with index \(\nu\le \beta-1=1.5>1\), but is generally not GMC. This setting directly corresponds to the short-range polynomial-dependence regime covered by Theorem 3.1. The revised data-generating process and dependence verification appear on page 11, in Example 1.
>
> We also added a paragraph titled “Verification of the moment and temporal-dependence conditions” at the beginning of Section 4. It states that Example 1 uses a polynomially decaying linear process satisfying the DAN(2) condition, whereas Examples 2 and 3 use VAR(1) processes satisfying GMC-type dependence. Thus, the revised simulations now cover both the polynomial-dependence and geometrically dependent settings addressed by the theory. This verification appears on page 10.

---

> > ### Comment · Reviewer_voTS · 2026-06-03
> >
> > The authors have adequately addressed the concern about moment conditions and simulation distributions. No further action is needed on this specific issue.

---

> ### Author Response · Authors · 2026-05-23
> **Response to Reviewer voTS, Part 2/3**
>
> **Response to Reviewer voTS, Part 2/3**
>
> ### Response to Requested Change 3
>
> We thank the reviewer for raising this important point. We agree that the choice of block length reflects a central trade-off between efficiency and robustness: longer blocks improve the precision of each block-wise estimate, whereas shorter blocks increase the robustness gained from the median aggregation. We have revised the manuscript and added new supplementary simulations to address this issue more explicitly.
>
> First, on the theoretical side, the revised results provide a principled logarithmic choice for the number of blocks. Specifically, Theorems 3.2 and 3.3 take
>
> $$
> k=\lceil C_0\log(n\vee d)\rceil,
> $$
>
> with feasibility conditions \(3\le k\le n/2\) for mean estimation and \(3\le k\le (n-l)/2\) for lag-\(l\) autocovariance estimation. Equivalently, the block length is \(m=\lfloor n/k\rfloor\). Under the short-range dependence regimes considered in this paper, the dependence strength enters the constants in the concentration bounds, but not the leading rate or the logarithmic order of \(k\). We have therefore used \(k=\lceil 2\log(n\vee d)\rceil\) as a practical default in the numerical experiments. These revisions appear on pages 8--9, in Theorems 3.2--3.3 and the surrounding discussion.
>
> Second, to directly assess the stability of the method with respect to \(k\), we have added a new sensitivity analysis in Example 5 of the supplementary material. This experiment varies \(k\) for both mean and autocovariance estimation under Normal, Student’s \(t\), and Pareto distributions, and across several levels of temporal dependence. The results make the robustness--efficiency trade-off explicit. At one extreme, \(k=1\), which reduces the MoM estimator to the sample mean, is competitive under light-tailed data but can be substantially worse under heavy-tailed autocovariance estimation. In contrast, the logarithmic rule \(k=\lceil 2\log(n\vee d)\rceil\) is consistently close to the error-minimizing choice across different tail behaviors, dependence strengths, and sample sizes. The error curves are also relatively flat around the optimum, indicating that the method is stable to moderate changes in \(k\).
>
> Finally, we have added discussion clarifying that, under stationarity, the block means themselves remain unbiased; the practical trade-off is therefore mainly between the variance of each block-wise estimate and the robustness/stability of the median aggregation. We also note that this trade-off may become more delicate under long-memory dependence, and that a fully adaptive choice of \(k\) for unknown tail and dependence strengths is an interesting direction for future work.
>
> ### Response to Requested Change 4
>
> We thank the reviewer for this helpful comment. We agree that a practical guideline for choosing the number of blocks \(k\) should be provided more explicitly. We have revised the manuscript and added a supplementary sensitivity analysis to address this issue.
>
> First, in the revised theoretical results, the block choice has been simplified. Theorems 3.2 and 3.3 now use the logarithmic rule
>
> $$
> k=\lceil C_0\log(n\vee d)\rceil,
> $$
>
> with the feasibility condition \(3\le k\le n/2\) for mean estimation and \(3\le k\le (n-l)/2\) for lag-\(l\) autocovariance estimation. Thus, the previous more complicated admissible ranges have been removed. These revisions appear on pages 8--9, in Theorems 3.2--3.3 and the surrounding discussion.
>
> Second, for implementation, we now recommend the concrete default choice
>
> $$
> k=\lceil 2\log(n\vee d)\rceil,
> $$
>
> and use this rule throughout Examples 1--3, rather than selecting \(k\) by hand. This provides a simple and reproducible practical guideline that is aligned with the theoretical logarithmic order. The use of this default rule is stated in the simulation descriptions on pages 11, 13, and 15.
>
> Third, following the reviewer’s suggestion, we have added a new sensitivity analysis in Example 5 of the supplementary material. This experiment varies \(k\) across different tail behaviors, dependence strengths, and sample sizes for both mean and autocovariance estimation. The results show that the default rule \(k=\lceil 2\log(n\vee d)\rceil\) is consistently close to the error-minimizing choice, and that the error curves are relatively flat over a broad range around the optimum. This indicates that the proposed estimator is stable to moderate changes in \(k\). Thus, the revised manuscript now provides both a practical default rule and direct sensitivity evidence supporting its use.

---

> > ### Comment · Reviewer_voTS · 2026-06-03
> > **Follow-up comment on the response to issues 3 and 4**
> >
> > Thank you for the revisions. The addition of a default rule \(k = \lceil 2\log(n\vee d)\rceil\) and the sensitivity analysis are helpful. The revised theorems now only state the very weak upper bound \(n/2\). Please clarify whether the theory has been strengthened to permit the weak upper bound, or whether the earlier \(O(\sqrt{n\log n})\) restriction is still implicitly needed.

---

> > > ### Author Response · Authors · 2026-06-04
> > >
> > > Thank you for pointing out this. We clarify that the revised theorems do not claim that every $k\le n/2$ yields the displayed optimal rate. In Theorems 3.2 and 3.3, $k$ is fixed at logarithmic order, $k=\lceil C_0\log(n\vee d)\rceil$. The conditions $3\le k\le n/2$ for mean and $3\le k\le (n-l)/2$ for lag-$l$ autocovariance are only feasibility conditions ensuring nondegenerate block sizes. The earlier $O(\sqrt{n\log(n\vee d)})$-type upper restriction is not needed in the revised proof. More generally, the proof gives an intermediate bound scaling as $\sqrt{k/n}$, so taking $k$ much larger than logarithmic order would not preserve the displayed optimal rate.

---

> > > > ### Comment · Reviewer_voTS · 2026-06-05
> > > >
> > > > Thank you for the clarification. No further action is needed on this point.

---

> ### Author Response · Authors · 2026-05-23
> **Response to Reviewer voTS, Part 3/3**
>
> **Response to Reviewer voTS, Part 3/3**
>
> ### Response to Requested Change 5
>
> We thank the reviewer for pointing this out. We have revised the manuscript to define these quantities explicitly.
>
> In the revised version, the coordinate-wise functional dependence measure is defined in Section 2.2 as
>
> $$
> \delta_{i,q,j}:=\|X_{i,j}-X'_{i,j}\|_q,
> $$
>
> where \(X'_i\) is the coupled version of \(X_i\). In Assumption 1, we now define
>
> $$
> \omega_{2,j}:=\|X_{i,j}\|_2=\left(\mathbb{E}\left|X_{i,j}\right|^2\right)^{1/2},
> \qquad
> \omega_2=\max_{j\in[d]}\omega_{2,j}.
> $$
>
> For autocovariance estimation, the analogous fourth-moment quantity is defined as
>
> $$
> \omega_{4,j}:=\|X_{i,j}\|_4=\left(\mathbb{E}\left|X_{i,j}\right|^4\right)^{1/4},
> \qquad
> \omega_4=\max_{j\in[d]}\omega_{4,j}.
> $$
>
> These definitions appear on page 5 in Section 2.2, on page 8 in Assumption 1, and on page 9 in Assumption 3.
>
> In addition, since the revised theory now requires only finite second moments for mean estimation and finite fourth moments for autocovariance estimation, the former third-moment quantities \(\omega_{3,j}\) and \(\delta_{i,3}\), as well as the sixth-moment quantities used in the earlier version, have been removed from the manuscript.
>
> ### Response to Requested Change 6
>
> We thank the reviewer for pointing this out. The typo has been corrected. The sentence now reads “Two commonly considered decay rates are ...” in Section 2.2. This correction appears on page 5, near the beginning of Section 2.2.
>
> ### Response to Requested Change 7
>
> We thank the reviewer for pointing out this inconsistency. We have checked and corrected the labels across the text, tables, and figures.
>
> In the revised manuscript, the distributions are now stated consistently. The mean-estimation examples use the standardized Student’s \(t_3\) distribution, which satisfies the finite-second-moment condition required by the revised mean-estimation theory. In contrast, the autocovariance example uses the standardized Student’s \(t_5\) distribution, together with a Pareto distribution with shape parameter \(4.5\), both of which have finite fourth moments as required by Theorem 3.3. The revised distributional specifications appear on page 13 for Example 2 and page 14 for Example 3.
>
> We again thank the reviewer for the thoughtful evaluation and constructive suggestions, which helped us improve the theoretical presentation, simulation design, and clarity of the manuscript.

---

> > ### Comment · Reviewer_voTS · 2026-06-03
> >
> > The authors have adequately addressed all three issues. No further action is required on these points.

---

### Decision · Action_Editor_9zUd · 2026-06-18

**Recommendation:** Accept as is

**Audience:**

Yes, researcher in high-dimensional inference of mean and covariance matrix would be interested in the contribution.

**Claims And Evidence:**

The paper made an interesting contribution to the area of high-diemensional statsitics. Though lack of accuracy in its first submision, all are corrected in the revision procecess. The two reviewers carefully reviewed the papers and make concrete comments to the paper. All were well addressed by the authors. The two reviewers are now satisfied with its current form. I concur with the them and think the current form is accurate, convincing and clearly evidenced.